

Biogeosciences  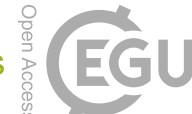

# Dissolved CH$_4$ coupled to photosynthetic picoeukaryotes in oxic waters and to cumulative chlorophyll *a* in anoxic waters of reservoirs

**Elizabeth León-Palmero**[1], **Alba Contreras-Ruiz**[1], **Ana Sierra**[2], **Rafael Morales-Baquero**[1], and **Isabel Reche**[1,3]

[1]Instituto del Agua, Departamento de Ecología, Universidad de Granada, 18071, Granada, Spain
[2]Instituto Universitario de Investigación Marina (INMAR), Departamento de Química Física,
Facultad de Ciencias del Mar y Ambientales, Universidad de Cádiz, Puerto Real, 11510, Cádiz, Spain
[3]Research Unit "Modeling Nature" (MNat), Universidad de Granada, 18071, Granada, Spain

**Correspondence:** Isabel Reche (ireche@ugr.es)

**Abstract.** Methane (CH$_4$) emissions from reservoirs are responsible for most of the atmospheric climatic forcing of these aquatic ecosystems, comparable to emissions from paddies or biomass burning. Primarily, CH$_4$ is produced during the anaerobic mineralization of organic carbon in anoxic sediments by methanogenic archaea. However, the origin of the recurrent and ubiquitous CH$_4$ supersaturation in oxic waters (i.e., the methane paradox) is still controversial. Here, we determined the dissolved CH$_4$ concentration in the water column of 12 reservoirs during summer stratification and winter mixing to explore CH$_4$ sources in oxic waters. Reservoir sizes ranged from 1.18 to 26.13 km$^2$. We found that dissolved CH$_4$ in the water column varied by up to 4 orders of magnitude (0.02–213.64 µM), and all oxic depths were consistently supersaturated in both periods. Phytoplanktonic sources appear to determine the concentration of CH$_4$ in these reservoirs primarily. In anoxic waters, the depth-cumulative chlorophyll *a* concentration, a proxy for the phytoplanktonic biomass exported to sediments, was correlated to CH$_4$ concentration. In oxic waters, the photosynthetic picoeukaryotes' abundance was significantly correlated to the dissolved CH$_4$ concentration during both the stratification and the mixing. The mean depth of the reservoirs, as a surrogate of the vertical CH$_4$ transport from sediment to the oxic waters, also contributed notably to the CH$_4$ concentration in oxic waters. Our findings suggest that photosynthetic picoeukaryotes can play a significant role in determining CH$_4$ concentration in oxic waters, although their role as CH$_4$ sources to explain the methane paradox has been poorly explored.

## 1 Introduction

Lakes and reservoirs are significant sources of methane (CH$_4$), affecting the atmospheric climatic forcing (Deemer et al., 2016). The estimated contribution of lakes to the global emission budget is ca. 71.6 Tg CH$_4$ yr$^{-1}$ (Bastviken et al., 2011), and the specific contribution of reservoirs ranges between 4 and 70 Tg CH$_4$ yr$^{-1}$, representing up to 10 % of total CH$_4$ emissions (Deemer et al., 2016). Although freshwater only covers about 5 %–8 % of the Earth's surface (Mitsch et al., 2012), it emits more CH$_4$ than the ocean surface (Saunois et al., 2016). Traditionally, the net CH$_4$ production is determined by archaeal methanogenesis, which produces methane as an end product of organic matter degradation in anoxic conditions, and to methanotrophs, which consume it in oxic conditions (Schubert and Wehrli, 2018). In freshwater ecosystems, the anoxic sediments are a primary source of CH$_4$ (Segers, 1998), where methanogens are very sensitive to temperature and quantity and quality of the organic matter used as substrate (Marotta et al., 2014; Rasilo et al., 2015; Sepulveda-Jauregui et al., 2018; Thanh-Duc et al., 2010; West et al., 2012; Yvon-Durocher et al., 2014). They are also affected by the extent of anoxia in the sediments insomuch as they are obligate anaerobes and will not survive

and produce CH$_4$ under aerobic conditions (Chistoserdova et al., 1998; Schubert and Wehrli, 2018). However, many observations from freshwater and marine water have detected CH$_4$ supersaturation in the oxic layers, a widespread phenomenon described as the "methane paradox" (Bogard et al., 2014; Damm et al., 2010; Donis et al., 2017; Grossart et al., 2011; Kiene, 1991; Murase et al., 2003; Owens et al., 1991; Schmidt and Conrad, 1993; Schulz et al., 2001; Tang et al., 2014, 2016).

This persistent CH$_4$ supersaturation in oxic layers of marine and freshwater ecosystems requires extra inputs to compensate for the CH$_4$ losses by methanotrophy and the emissions toward the atmosphere. CH$_4$ inputs may come from anoxic sediments or from in situ sources in the oxic layers. The transport of CH$_4$ from the bottom and littoral sediments in shallow zones has been proposed to explain the supersaturation in the surface waters of some lakes (Bastviken et al., 2004; Encinas Fernández et al., 2016; Michmerhuizen et al., 1996; Murase et al., 2003; Peeters et al., 2019; Rudd and Hamilton, 1978). The vertical transport may be relevant in small lakes, but in deep and thermally stratified systems, the vertical diffusion rates of dissolved gases across the thermocline are too low, and there is no apparent CH$_4$ upward movement from the hypolimnion (Peeters et al., 1996; Rudd and Hamilton, 1978). In fact, Thalasso et al. (2020) determined that there was no exchange between the hypolimnion and the epilimnion in a Siberian lake. The CH$_4$ produced in the sediments and the hypolimnion was assimilated there. Consequently, the CH$_4$ in the epilimnion came from lateral transport and in situ production. Lateral CH$_4$ transport from shallow sediments of the littoral zones may be a significant source in the open surface of some lakes and reservoirs. DelSontro et al. (2018) found that CH$_4$ transport from littoral zones was relevant for the dissolved CH$_4$ in the epilimnion of small lakes. However, lateral transport does not fully explain CH$_4$ supersaturation in the open ocean, and large freshwater ecosystems, and, hence, other in situ CH$_4$ sources, likely occur (Damm et al., 2010; DelSontro et al., 2018; Grossart et al., 2011; Khatun et al., 2020; Owens et al., 1991; Schmidt and Conrad, 1993; Schulz et al., 2001; Scranton and Brewer, 1977; Tang et al., 2014; Tilbrook and Karl, 1995).

Previous works demonstrated the in situ CH$_4$ production in oxic waters using stable isotope techniques in experiments, mesocosms, and field samples (Bižić et al., 2020; Bogard et al., 2014; DelSontro et al., 2018; Hartmann et al., 2020; Tang et al., 2016) and using molecular approaches (Grossart et al., 2011; Khatun et al., 2020; Yao et al., 2016a). In the literature, there are different alternatives proposed as CH$_4$ sources. On the one hand, there is the occurrence of methanogenesis in micro-anoxic niches in the guts of zooplankton and within sinking particles (de Angelis and Lee, 1994; Karl and Tilbrook, 1994). In both micro-anoxic niches, the CH$_4$ production appeared to be too low to sustain the total CH$_4$ supersaturation of the oxic waters (Schmale et al., 2018; Tang et al., 2014). On the other hand, there is a consistent link between dissolved CH$_4$ concentration and autotrophic organisms, primary production, and chlorophyll $a$ concentration (Bogard et al., 2014; Grossart et al., 2011; Owens et al., 1991; Schmidt and Conrad, 1993; Tang et al., 2014). Grossart et al. (2011) detected potential methanogenic CE3 *Archaea* attached to photoautotrophs as *Chlorophyta* (*Eukarya*) and cyanobacteria (*Bacteria*) in the epilimnion of an oligotrophic lake and confirmed the production of CH$_4$ in the presence of oxygen in laboratory incubations. If occurring, that symbiosis would require that the methanogenic microorganisms tolerate the oxygen exposure, as has been observed by several authors (Angel et al., 2011, 2017; Jarrell, 1985), in contrast to general belief. New findings suggest that the link between phytoplankton and dissolved CH$_4$ may rely on diverse metabolic pathways in *Bacteria* CE4 and *Eukarya*. These metabolic pathways contribute to the dissolved CH$_4$ in oxic waters due to the degradation of methylated compounds. In the open ocean, archaea and bacteria appear to metabolize the algal osmolyte dimethylsulfoniopropionate, producing methane as a by-product (Damm et al., 2008, 2010, 2015; Zindler et al., 2013). Common methyl-containing substances like methionine produce methane in algae, saprotrophic fungi, and plants (Lenhart et al., 2012, 2015, 2016). Another reported pathway is the degradation of methylphosphonates (MPn's CE5) as an alternative source of phosphorus (P) in phosphate-starved bacterioplankton. The hydrolysis of these compounds, using the enzyme C–P lyase, also releases methane as a by-product. This pathway appears in chronically P-starved ecosystems as the ocean gyres, oligotrophic lakes, and microbial mats (Beversdorf et al., 2010; Carini et al., 2014; Gomez-Garcia et al., 2011; Karl et al., 2008; Repeta et al., 2016; Teikari et al., 2018; del Valle and Karl, 2014; Wang et al., 2017; Yao et al., 2016a). Recent studies using phytoplankton cultures and stable isotope techniques propose that the production of CH$_4$ may rely directly on the photoautotrophic carbon fixation of algae and cyanobacteria (Bižić et al., 2020; Hartmann et al., 2020; Klintzsch et al., 2019; Lenhart et al., 2016). These sources of CH$_4$ in oxic waters, however, still have not been tested simultaneously in reservoirs, despite the known high contribution of these freshwater ecosystems to global CH$_4$ emissions.

In this study, we measured the dissolved CH$_4$ concentration in the water column of 12 reservoirs that cover a broad spectrum of sizes, ages, morphometries, and trophic states during the summer stratification and winter mixing (León-Palmero et al., 2020). Our objective was to assess the relative contribution of different sources of CH$_4$ in the oxic waters and to shed light on the methane paradox depending on reservoir properties. We explored the following CH$_4$ sources in oxic waters: (1) vertical and lateral transport of CH$_4$ from hypolimnetic and littoral waters, (2) in situ production by methanogenic *Archaea* tolerant to oxygen, (3) in situ production by methylphosphonate degradation, and (4) in situ production by photosynthetic microorganisms. We used the concentration chlorophyll $a$, the primary productivity, and the

abundance of photosynthetic picoeukaryotes and cyanobacteria as variables for the photosynthetic signatures. The photosynthetic picoeukaryotes are a relevant part of the freshwater phytoplankton, but their role in the methane paradox has been particularly little studied.

## 2 Methods

### 2.1 Studied reservoirs, morphometry, and vertical profiles

We sampled 12 reservoirs located in southern Spain (Fig. 1) between July 2016 and August 2017 once during the summer stratification and once during winter mixing. In Table 1, we show the geographical coordinates, age, and the morphometry description of the studied reservoirs. The reservoirs were built between 1932 and 2003, for water supply and agriculture irrigation, and they are located in watersheds with different lithologies and land uses (more details can be found in León-Palmero et al., 2019, 2020). These reservoirs differ in morphometric, chemical, and trophic characteristics, covering a wide range of concentrations of dissolved organic carbon (DOC), total nitrogen (TN), total phosphorus (TP), and chlorophyll $a$ (Table 2). All raw data for the water column were deposited in the PANGAEA database (https://doi.org/10.1594/PANGAEA.912535, last access: TS1).

We obtained the reservoir surface area, perimeter, and volume using the following open databases: Infraestructura de Datos Espaciales de Andalucía (IDEAndalucia; http://www.ideandalucia.es/portal/web/ideandalucia/, last access: TS2) and the Ministerio para la Transición Ecológica (https://www.embalses.net/, last access: TS3).

The mean depth was calculated as follows (Eq. 1):

$$\text{Mean depth (m)} = \frac{\text{Volume } (\text{m}^3)}{\text{Surface area } (\text{m}^2)}. \qquad (1)$$

The shoreline development ratio ($D_L$) (Aronow, 1982) is a comparative index relating the shoreline length (i.e., the perimeter of the reservoir) to the circumference of a circle that has the same area. The closer this ratio is to 1, the more circular the lake. A large ratio ($\gg 1$) indicates that the shoreline is more scalloped than a low ratio. The equation is as follows (Eq. 2):

$$D_L = \frac{\text{Length of the shoreline (m)}}{2\sqrt{\pi \text{ Area } (\text{m}^2)}}. \qquad (2)$$

The shallowness index (m$^{-1}$) was obtained by dividing the shoreline development index ($D_L$) by the mean depth (m), as in Eq. (3):

$$\text{Shallowness index } \left(\text{m}^{-1}\right) = \frac{D_L}{\text{Mean depth (m)}}. \qquad (3)$$

We sampled the water column near the dam, in the open water of the reservoir. During the stratification and the mixing period, we selected the same location. First, we performed a vertical profile of the reservoir using a Sea-Bird 19plus CTD profiler, coupled to a Spherical Underwater Quantum Sensor (LI-193R), and a fluorimeter Turner® SCUFA (model CYCLOPS-7) for continuous measurements of temperature (°C), dissolved oxygen (μM), conductivity (μS cm$^{-1}$ TS4), turbidity (FTU – formazin turbidity unit), density (kg m$^{-3}$), photosynthetic active radiation, chlorophyll $a$ fluorescence (μg L$^{-1}$), specific conductance (μS cm$^{-1}$), and salinity (psu – practical salinity units). Then, based on the temperature and oxygen profiles, we selected from six to nine depths, representative of the oxic and anoxic layers and the transition between them in the different reservoirs. We took the water samples using a UWITEC sampling bottle of 5 L with a self-closing mechanism. We collected samples for the dissolved CH$_4$ analysis in 125 or 250 mL airtight Winkler bottles by duplicate CE6 (250 mL) or triplicate (125 mL). We filled up the bottles very carefully from the bottom to avoid the formation of bubbles and minimize the loss of CH$_4$ during field sampling. We preserved the samples with a solution of HgCl$_2$ (final concentration 1 mM CE7) to inhibit biological activity and sealed the bottles with Apiezon® grease to prevent gas exchanges. We also took samples from each depth from the chemical and biological analysis explained below. We also measured barometric pressure using a multi-parameter probe (Hanna HI 9828) for the gas saturation calculations. We calculated the saturation values (%) for dissolved oxygen as the ratio of the dissolved gas measured and the gas concentration expected in equilibrium. We calculated the gas concentration in equilibrium, taking into account the differences in temperature, salinity, and barometric pressure (Mortimer, 1956).

### 2.2 Dissolved CH$_4$ in the water column

We stored the Winkler bottles in the dark at room temperature until analysis in the laboratory. We measured dissolved CH$_4$ using headspace equilibration in a 50 mL airtight glass syringe (Agilent P/N 5190–1547) (Sierra et al., 2017). We obtained two replicates for each 150 mL Winkler bottle and three replicates for each 250 mL Winkler bottle. We took a quantity of 25 g of water ($\pm 0.01$ g) using the airtight syringe and added a quantity of 25 mL of a standard gas mixture that had a methane concentration similar to atmospheric values (1.8 ppmv) to complete the volume of the syringe. The syringes were shaken for 5 min (Vibromatic, Selecta) to ensure mixing, and we waited 5 min to reach complete equilibrium. Then, the gas in the syringe was injected manually into the gas chromatograph (GC; Bruker® GC-450) equipped with a hydrogen flame ionization detector (FID). We calibrated the detectors daily using three standard gas mixtures with CH$_4$ mixing ratios of 1952, 10 064, and 103 829 ppbv, made and certified by Air Liquide (France). We calculated the gas concentration in the water samples

**Table 1.** Geographical location and morphometric description of the studied reservoirs.

| Reservoir | Latitude (°, decimal degrees) | Longitude (°, decimal degrees) | Altitude (m) | Construction year | Reservoir area (km$^2$) | Reservoir capacity (hm$^3$) | Mean depth (m) | Shoreline development index ($D_L$) | Shallowness index (m$^{-1}$) |
|---|---|---|---|---|---|---|---|---|---|
| Cubillas | 37.27 | −3.68 | 640 | 1956 | 1.94 | 18.74 | 9.66 | 2.00 | 0.21 |
| Colomera | 37.40 | −3.72 | 810 | 1990 | 2.76 | 40.18 | 14.56 | 3.35 | 0.23 |
| Negratín | 37.56 | −2.95 | 618 | 1984 | 23.51 | 567.12 | 24.12 | 5.90 | 0.24 |
| La Bolera | 37.76 | −2.90 | 950 | 1967 | 2.89 | 53.19 | 18.40 | 4.05 | 0.22 |
| Los Bermejales | 36.99 | −3.89 | 852 | 1958 | 5.95 | 103.12 | 17.33 | 2.90 | 0.17 |
| Iznájar | 37.26 | −4.33 | 425 | 1969 | 26.13 | 981.12 | 37.55 | 5.76 | 0.15 |
| Francisco Abellán | 37.31 | −3.27 | 942 | 1991 | 2.43 | 58.21 | 23.95 | 3.80 | 0.16 |
| Béznar | 36.92 | −3.55 | 486 | 1986 | 1.60 | 52.90 | 33.06 | 2.65 | 0.08 |
| San Clemente | 37.86 | −2.65 | 1050 | 1990 | 3.76 | 117.92 | 31.36 | 3.43 | 0.11 |
| El Portillo | 37.81 | −2.79 | 920 | 1999 | 1.18 | 32.90 | 27.88 | 3.69 | 0.13 |
| Jándula | 38.23 | −3.97 | 350 | 1932 | 8.43 | 321.99 | 38.20 | 7.10 | 0.19 |
| Rules | 36.86 | −3.49 | 239 | 2003 | 3.06 | 110.78 | 36.20 | 3.09 | 0.09 |

**Table 2.** Sampling date; depth of the mixing layer (m); mean values of the DOC, TN, and TP concentrations; DIN : TP ratio; and chlorophyll *a* concentration in the water column of the studied reservoirs during the stratification and the mixing period. The depth of the mixing layer was inferred from the temperature profile.

| Reservoir | Period | Sampling date | DOC (µM C) | TN (µM N) | TP (µM P) | DIN : TP (µmol N : µmol P) | Chl *a* (µg L$^{-1}$) |
|---|---|---|---|---|---|---|---|
| Cubillas | Stratification | 15 Jul 2016 | 172.1 | 60.4 | 1.84 | 26 | 17.8 |
| | Mixing | 6 Feb 2017 | 240.5 | 97.4 | 0.78 | 111 | 8.4 |
| Colomera | Stratification | 22 Jul 2016 | 99.4 | 181.4 | 0.78 | 240 | 2.1 |
| | Mixing | 7 Mar 2017 | 123.3 | 112.5 | 0.44 | 292 | 0.5 |
| Negratín | Stratification | 27 Jun 2016 | 109.7 | 21.2 | 0.80 | 28 | 1.2 |
| | Mixing | 16 Feb 2017 | 148.9 | 19.7 | 0.24 | 65 | 7.7 |
| La Bolera | Stratification | 28 Jun 2016 | 123.7 | 17.3 | 0.61 | 25 | 2.0 |
| | Mixing | 8 Apr 2017 | 107.4 | 34.4 | 0.15 | 178 | 0.8 |
| Los Bermejales | Stratification | 7 Sep 2016 | 94.2 | 30.4 | 0.42 | 65 | 1.8 |
| | Mixing | 17 Mar 2017 | 101.5 | 30.6 | 0.31 | 89 | 13.1 |
| Iznájar | Stratification | 9 Sep 2016 | 116.8 | 278.5 | 0.39 | 729 | 5.1 |
| | Mixing | 15 Mar 2017 | 147.5 | 260.0 | 1.16 | 393 | 1.1 |
| Francisco Abellán | Stratification | 28 Sep 2016 | 90.6 | 27.8 | 0.28 | 200 | 1.9 |
| | Mixing | 21 Mar 2017 | 118.0 | 28.5 | 0.47 | 63 | 1.1 |
| Béznar | Stratification | 7 Oct 2016 | 74.3 | 74.2 | 0.68 | 227 | 6.0 |
| | Mixing | 23 Feb 2017 | 121.6 | 105.6 | 0.95 | 104 | 3.7 |
| San Clemente | Stratification | 17 Jul 2017 | 104.1 | 32.0 | 0.39 | 65 | 3.5 |
| | Mixing | 28 Mar 017 | 119.4 | 35.9 | 0.21 | 145 | 1.1 |
| El Portillo | Stratification | 18 Jul 2017 | 78.0 | 22.8 | 0.17 | 102 | 2.4 |
| | Mixing | 30 Mar 2017 | 76.4 | 34.4 | 0.26 | 109 | 1.7 |
| Jándula | Stratification | 24 Jul 2017 | 359.9 | 34.3 | 0.78 | 43 | 2.3 |
| | Mixing | 5 Apr 2017 | 399.4 | 46.2 | 0.37 | 104 | 1.2 |
| Rules | Stratification | 10 Jul 2017 | 81.2 | 23.2 | 0.21 | 83 | 3.7 |
| | Mixing | 7 Apr 2017 | 68.5 | 38.0 | 0.43 | 142 | 3.3 |

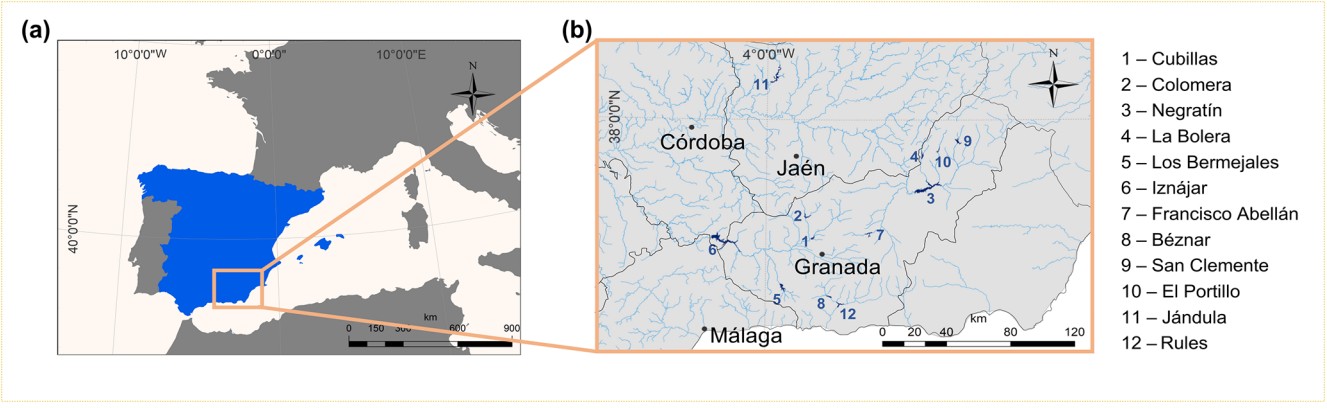

**Figure 1.** Geographical location of the studied reservoirs. **(a)** The location area of the studied reservoirs delimited by an orange box in the south of the Iberian Peninsula. **(b)** Detailed location of the 12 reservoirs with the numbers (1–12) and their corresponding names listed on the side. Geographical coordinates appear in Table 1. We obtained these maps using ArcGIS$^®$ 10.2 software (ESRI, 2012) under the Universidad de Granada license. ©ESRI: ArcGIS, Redlands, CA.

from the concentration measured in the headspace using the Bunsen functions for CH$_4$ (Yamamoto et al., 1976; Wiesenburg and Guinasso, 1979). The precision in the quantification of the gas mixture of CH$_4$ used in the headspace equilibrium (1.8 ppmv) expressed as the coefficient of variation was 3.7 % ($n = 123$). The precision of the measurement of the dissolved CH$_4$ concentration, which included the analytical processing of the samples and the equilibration step, was 3.6 % for four to six replicates of each sample. We calculated the saturation values (%) as the ratio of the concentration of the dissolved gas measured to the gas concentration expected in equilibrium considering the temperature, salinity, and barometric pressure of each reservoir. We used the atmospheric gas concentrations provided by the Global Greenhouse Gas Reference Network website (https://www.esrl.noaa.gov/gmd/ccgg/index.html, last access: TS5), which is part of the National Oceanic and Atmospheric Administration (NOAA) Earth System Research Laboratory in Boulder, Colorado. We calculated the 2016 global mean atmospheric concentrations for CH$_4$ (Dlugokencky, 2019) from the 2016 global monthly mean. The differences among these values and the local atmospheric concentrations are assumed to be small compared with the high dissolved concentrations obtained in the studied reservoirs.

## 2.3 Chemical analysis in the water column

From the discrete sampling, we selected thee or four representative depths of the epilimnion, metalimnion (oxycline), and hypolimnion–bottom CE8 layers for nutrient analysis during the stratification period. We also selected three or four equivalent depths during the mixing period. In total, we analyzed 77 samples: 41 samples from the stratification period and 36 samples from the mixing period. We determined total nutrients using unfiltered water, while we filtered the samples through pre-combusted 0.7 μm pore-size Whatman

GF/F glass-fiber filters for the dissolved nutrients. We acidified the samples for dissolved organic carbon (DOC), total dissolved nitrogen (TDN), and total nitrogen (TN) with phosphoric acid (final pH <2). We measured DOC, TN, and TDN by high-temperature catalytic oxidation using a Shimadzu total organic carbon (TOC) analyzer (Model TOC-VCSH) coupled to a nitrogen analyzer (TNM-1). We calibrated the instrument using a four-point standard curve of dried potassium hydrogen phthalate for DOC and dried potassium nitrate for TN and TDN (Álvarez-Salgado and Miller, 1998). We analyzed two replicates and three to five injections per replicate for each sample. We purged the DOC samples with phosphoric acid for 20 min to eliminate all the dissolved inorganic carbon. The precision of the DOC measurements expressed as the mean coefficient of variation was 3.0 %. The mean precision for the TN and TDN was 8.2 % and 2.9 %, respectively.

We measured the NO$_3^-$ concentration by duplicate with the ultraviolet spectrophotometric method, using a Perkin Elmer UV Lambda 40 spectrophotometer at wavelengths of 220 nm and correcting for DOC absorbance at 275 nm (Baird et al., 2012). The mean coefficient of variation was 0.5 %. We measured NH$_4^+$ and NO$_2^-$ concentrations by inductively coupled plasma optical emission spectrometry (ICP-OES). Dissolved inorganic nitrogen (DIN) was calculated as the addition of the NO$_3^-$, NH$_4^+$, and NO$_2^-$ concentrations. The detection limits for the NH$_4^+$ and NO$_2^-$ concentrations were 3.6 and 1.4 μM, respectively. We measured total phosphorus (TP) concentration by triplicate using the molybdenum blue method (Murphy and Riley, 1962) after digestion with a mixture of potassium persulfate and boric acid at 120 °C for 30 min (Baird et al., 2012). The precision in the quantification of the TP concentration was 11.1 %.

Please note the remarks at the end of the manuscript.

## 2.4 Chlorophyll *a*, phytoplankton, and primary production in the water column

We determined the chlorophyll *a* concentration and the abundances of cyanobacteria and photosynthetic picoeukaryotes in all the depths sampled during the discrete samplings ($n = 178$). We determined the chlorophyll *a* concentration by filtering the particulate material of 500 to 2000 mL of water through pre-combusted Whatman GF/F glass-fiber filters. Then, we extracted the pigments from the filters with 95 % methanol in the dark at 4 °C for 24 h (Baird et al., 2012). We measured chlorophyll *a* (Chl *a*) absorption using a Perkin Elmer UV Lambda 40 spectrophotometer at the wavelength of 665 nm and for scattering correction at 750 nm. The detection limit was 0.1 µg L$^{-1}$.

To obtain the cumulative chlorophyll *a* in the whole water column (mg Chl *a* m$^{-2}$), from the discrete depths, we summed the concentration of Chl *a* from each stratum using the trapezoidal rule (León-Palmero et al., 2019), as indicated in Eq. (4):

$$\text{Cumulative Chl} a\ a = \sum_{k=1}^{n} X_{ik} \cdot \left( Z_{k+1} - \frac{Z_{k-1}}{2} \right), \tag{4}$$

where $Z$ stands for the depth considered, and $n$ is the number of depths sampled. $Z_k$ stands for the $n$ sampled depth; $X_{ij}$ is the Chl*a* *a* concentration (µg L$^{-1}$) at the depth $Z_k$.

We determined by triplicate the abundances of cyanobacteria and photosynthetic picoeukaryotes using flow cytometry using unfiltered water. We collected and fixed the samples with a mixture of 1 % paraformaldehyde and 0.05 % glutaraldehyde for 30 min in the dark at 4 °C. Then, we froze the samples in liquid nitrogen and stored them at −80 °C until analysis. We analyzed the samples in the FACSCalibur flow cytometer equipped with the BD CellQuest Pro software for data analysis. We used yellow–green 0.92 µm latex beads (Polysciences) as an internal standard to control the cytometer performance every day. We used different signals for groups determination: the side scatter (SSC), chlorophyll *a* (red fluorescence – FL3), phycoerythrin (orange fluorescence – FL2), and phycocyanin (blue fluorescence – FL4), following the protocols and indications for data analysis of previous works (Cellamare et al., 2010; Collier, 2000; Corzo et al., 1999; Gasol and Giorgio, 2000; Liu et al., 2014). In Fig. S13 in the Supplement, we show a cytogram of the populations of cyanobacteria and photosynthetic picoeukaryotes. The mean coefficient of variation for the abundances of cyanobacteria and photosynthetic picoeukaryotes was 8.8 % and 11.4 %, respectively.

We estimated gross primary production (GPP), net ecosystem production (NEP), and ecosystem respiration ($R$) by measuring temporal changes in dissolved oxygen concentration and temperature using a miniDOT (PME) submersible waterlogger during the stratification period. We recorded measurements every 10 min for 24–4 h CE9 during the same sampling days. Briefly, the equation for estimating free-water metabolism from measurements of dissolved oxygen was established by Odum (1956) (Eq. 5):

$$\Delta \text{O}_2 / \Delta t = \text{GPP} - R - F - A, \tag{5}$$

where $\Delta \text{O}_2 / \Delta t$ is the change in dissolved oxygen concentration through time, $F$ is the exchange of O$_2$ with the atmosphere, and $A$ is a term that combines all other processes that may cause changes in the dissolved oxygen concentration as horizontal or vertical advection, and it is often assumed to be negligible. The calculations were performed as in Staehr et al. (2010). The physical gas flux was modeled as follows (Eq. 6):

$$F \left( g\ \text{O}_2\ \text{m}^{-2}\,\text{h}^{-1} \right) = k \left( \text{O}_{2\,\text{meas}} - \text{O}_{2\,\text{sat}} \right), \tag{6}$$

where $F$ is the physical gas flux, and $k$ (m h$^{-1}$) is the piston velocity estimated following the equation of Jähne et al. (1987) and the indications of Staehr et al. (2010). O$_{2\,\text{meas}}$ is the actual oxygen concentration (mg mL$^{-1}$), and O$_{2\,\text{sat}}$ is the oxygen concentration in water in equilibrium with the atmosphere at ambient temperature and salinity.

We calculated the hourly net ecosystem production (NEP$_{\text{hr}}$) and the daytime net ecosystem production (NEP$_{\text{daytime}}$) following Eqs. (7) (Cole et al., 2000) and (8):

$$\text{NEP}_{\text{hr}} \left( g\ \text{O}_2\ \text{m}^{-3}\,\text{h}^{-1} \right) = \Delta \text{O}_2 \left( g\,\text{m}^{-3}\,\text{h}^{-1} \right) - F / Z_{\text{mix}}, \tag{7}$$

$$\text{NEP}_{\text{daytime}} \left( g\ \text{O}_2\ \text{m}^{-3}\,\text{daylight period}^{-1} \right)$$
$$= \text{mean NEP}_{\text{hr}} \text{ during daylight} \left( g\text{O}_2\text{m}^{-3}\,\text{h}^{-1} \right)$$
$$\times \text{Light hours (h)}. \tag{8}$$

NEP$_{\text{hr}}$ is directly derived from the changes in dissolved oxygen ($\Delta \text{O}_2$), after accounting for physical gas flux with the atmosphere ($F$). $Z_{\text{mix}}$ is the depth of the mixed layer (m), which was inferred from the temperature profile as the upper mixed zone where the temperature remains constant. NEP$_{\text{daytime}}$ is the portion of NEP between sunrise and sunset, when the photosynthesis takes place. We obtained the exact light hours from an online solar calculator (https://es.calcuworld.com/calendarios/calcular-salida-y-puesta-del-sol/, last access: TS6). We established the start and the end time for photosynthesis as 30 min before sunrise and 30 min after dawn (Schlesinger and Bernhardt, 2013). We obtained hourly $R$ ($R_{\text{hr}}$), $R$ during the daytime ($R_{\text{daytime}}$), and $R$ throughout the whole day ($R_{\text{day}}$), fol-

lowing Eqs. (9), (10), and (11), respectively:

$$R_{hr}\left(g\,O_2\,m^{-3}\,h^{-1}\right)$$
$$= \text{mean NEP}_{hr} \text{ during darkness}\left(g\,O_2\,m^{-3}\,h^{-1}\right), \quad (9)$$

$$R_{daytime}\left(g\,O_2\,m^{-3}\text{daylight period}^{-1}\right)$$
$$= R_{hr}\left(g\,O_2\,m^{-3}\,h^{-1}\right) \times \text{Light hours (h)}, \quad (10)$$

$$R_{day}\left(g\,O_2\,m^{-3}\,d^{-1}\right) = R_{hr}\left(g\,O_2\,m^{-3}\,h^{-1}\right) \times 24\,(h). \quad (11)$$

We calculated the respiration rate during the night (the period between 60 min after dawn and 60 min before sunrise) (Staehr et al., 2010), and we assumed that the respiration rate overnight was similar to the respiration rate throughout the day. Finally, we obtained the GPP and NEP for the day (Eqs. 12 and 13):

$$\text{GPP}\left(g\,O_2\,m^{-3}\,d^{-1}\right) = \text{NEP}_{daytime} + R_{daytime}, \quad (12)$$

$$\text{NEP}\left(g\,O_2\,m^{-3}\,d^{-1}\right) = \text{GPP} - R_{day}. \quad (13)$$

## 2.5 DNA analysis

We selected three or four representative depths for determining the abundance of the functional genes of the epilimnion, metalimnion (oxycline), and hypolimnion–bottom layers during the stratification period. We also selected three or four equivalent depths during the mixing period. In total, we analyzed 41 samples from the stratification period and 36 samples for the mixing period. We pre-filtered the water through 3.0 µm pore-size filters and extracted DNA following the procedure developed by Boström et al. (2004) for environmental samples. During the DNA extraction protocol, we combined a cell recovery step by centrifugation of 12–20 mL of the pre-filtered water, a cell lysis step with enzyme treatment (lysozyme and proteinase $K$), and, finally, the DNA recovery step with a co-precipitant (yeast tRNA) to improve the precipitation of low-concentration DNA. DNA was quantified using a DNA quantitation kit (Sigma-Aldrich) based on the fluorescent dye bisbenzimide (Hoechst 33258). Extracted DNA served as the template for PCR and quantitative PCR (qPCR) analysis to test the presence and abundance of the *mcrA* gene and the *phnJ* gene. For PCR analysis, we used the recombinant Taq DNA Polymerase (Thermo Fisher Scientific) using the Mastercycler X50 thermal cycler (Eppendorf). We ran the qPCR plates using SYBR Green as the reporter dye (PowerUp™ SYBR™ Green Master Mix, Thermo Fisher Scientific) in the Applied Biosystems 7500 Real-Time PCR System and the 7500 Software. In both cases, PCR and qPCR, we designed the standard reaction mix recipes and the thermocycling conditions using the provider specifications and primer requirements. We chose specific primers from studies performed in natural samples of freshwater. We used pure cultures as positive controls (more details below).

We targeted the alpha subunit of methyl-coenzyme reductase (*mcrA*) as a genetic marker to determine the existence and abundance of methanogenic *Archaea* in our samples. This gene appears to be an excellent marker, since all known methanogens have the methyl-coenzyme M reductase, which is the enzyme responsible for the conversion of a methyl group to CH$_4$ (Grabarse et al., 2001). We used specific primers from West et al. (2012), adapting their procedure. The forward primer was mcrAqF (5'-AYGGTATGGARCAGTACGA-3'), the reverse primer was mcrAqF (5'-TGVAGRTCGTABCCGWAGAA-3'), and the annealing temperature was 54 °C. The expected size of the PCR product was $\sim 200$ bp (bp – base pair) CE10. We used a culture of *Methanosarcina acetivorans* (ATCC 35395) as a positive control. We tested all the samples ($n = 77$). We also tested the presence of the *phnJ* gene, which encodes a subunit of the C–P lyase complex (Seweryn et al., 2015; White and Metcalf, 2007). This enzyme cleaves C–P bonds in phosphonate compounds, releasing methane, and changes in response to the phosphate availability (Yao et al., 2016a). We ran the amplification with a pair of primers previously used by Fox et al. (2014) and Yao et al. (2016a). The forward primer was PhnJoc1 (5'-AARGTRATMGAYCARGG-3'), and the reverse was PhnJoc2 (5'-CATYTTYGGATTRTCRAA-3'), adapting the PCR procedure from Yao et al. (2016a). The annealing temperature was 52.5 °C, and the positive controls were run using a pure culture of *Rhodopseudomonas palustris* (ATCC 33872). The expected size of the PCR product was $\sim 400$ bp. We checked the result of the amplification by running 1.5 % ($w/v$) agarose gel electrophoresis. If we did not detect amplification in the PCR or qPCR samples, we changed the standard procedure by increasing the DNA amount and the primers' concentration to corroborate the negative results. We tested all the samples ($n = 77$).

## 2.6 Statistical tests

We conducted all the statistical analysis in R (R Core Team, 2014), using the packages "car" (Fox and Weisberg, 2011), "nortest" (Gross and Ligges, 2015), and "mgcv" (Wood, 2011). We performed the Shapiro–Wilk test of normality analysis and Levene's test for homogeneity of variance across groups. We performed a one-way analysis-of-variance test (ANOVA) when the data were normally distributed. In case the data did not meet the assumptions of normality, we used the paired Kruskal–Wallis rank-sum (K–W) or Wilcoxon ($V$) tests. We analyzed the potential sources of dissolved CH$_4$ using simple regression analysis and generalized additive models (GAMs) (Wood, 2006). A GAM is a generalized model with a linear predictor involving a sum of smooth functions of covariates (Hastie and Tibshirani, 1986, 1990). The model structure is shown in Eq. (4):

$$y_i = f_1(x_{1i}) + f_2(x_{2i}) + \ldots + f_n(x_{ni}) + \in_i, \quad (4)$$

where $f_j$ is the smooth functions, and $\in_i$ is independent identically distributed $N\,(0, \sigma^2)$ random variables. We fit smoothing functions by penalized cubic regression splines. The cross-validation method (generalized cross-validation – GCV – criterion) estimates the smoothness of the functions. We fitted the models to minimize the Akaike information criterion (AIC) and the GCV values. We calculated the percentage of variance explained by the model (adjusted $R^2$) and the quality of the fit (deviance explained). We also fixed the effect of each predictor to assess the contribution of the other predictor on the total deviance explained. Then, the sum of the deviance explained by two predictors can be different from the deviance explained by the model due to interactive effects.

## 3 Results and discussion

### 3.1 Profile description

We found pronounced differences in the concentration of dissolved CH₄ of the studied reservoirs among depths and seasonal periods (Figs. 2–4 and S1–9). The concentration of dissolved CH₄ ranged up to 4 orders of magnitude, from 0.06 to 213.64 μM, during the summer stratification ($n = 96$), and it was less variable during the winter mixing ($n = 84$), ranging only from 0.02 to 0.69 μM. All depths were consistently supersaturated in CH₄, during both the stratification and mixing period (Table S1 in the Supplement). The dissolved CH₄ concentration and the percentage of saturation values were significantly higher during the stratification period than during the mixing period ($V = 78$, $p$ value < 0.001, and $V = 78$, $p$ value < 0.001, respectively). These differences in the concentration of dissolved CH₄ are coherent with the differences found in the CH₄ emissions from these reservoirs in the stratification and mixing periods (León-Palmero et al., 2020). The wide range in CH₄ concentrations found in this study covers values reported in temperate lakes (Donis et al., 2017; Grossart et al., 2011; Tang et al., 2014; West et al., 2016), to those found in tropical lakes and reservoirs (Murase et al., 2003; Naqvi et al., 2018; Okuku et al., 2019; Roland et al., 2017). In the surface mixing layer during the stratification period (i.e., epilimnion), we found values from 0.06 to 8.18 μM (Table S1), which is about 80 times the maximum values found in the surface waters of Lake Kivu (Africa) by Roland et al. (2017) and similar to the concentrations reported in subtropical and tropical reservoirs (Musenze et al., 2014, and references therein).

The dissolved CH₄ profiles showed considerable differences among depths during the summer stratification (Figs. 2a–4a and S1a–9a) but were very homogeneous during the winter mixing in all the reservoirs (Figs. 2b–4b and S1b–9b) (Table S1). Based on the differences found during the stratification period in the dissolved CH₄ profiles, we sorted the reservoirs into three types. The first type of CH₄ pro-

file included six reservoirs that were characterized by an increase in the dissolved CH₄ from the oxycline to the anoxic bottom, just above the sediments, where CH₄ concentration reached its maximum. In these reservoirs, the oxycline may be spatially coupled to the thermocline or not. When the oxycline and the thermocline were spatially coupled, the dissolved CH₄ concentration increased exponentially from the thermocline along the anoxic hypolimnion to the sediments. The reservoirs Béznar, San Clemente, and Iznájar showed this type of profile (Figs. 2a, S1a, and S2a). The existence of a sizeable almost-anoxic hypolimnion led to a massive accumulation of CH₄ in this layer. The differences in the CH₄ concentration between the surface and bottom waters were up to 3 orders of magnitude, as we found in Béznar (from the 0.25 to 56.17 μM; Fig. 2a), San Clemente (from the 0.23 to 45.15 μM; Fig. 1a), and Iznájar (from the 0.82 to 213.64 μM; Fig. S2a). When the oxycline and the thermocline were not spatially coupled, the dissolved CH₄ concentration increased just above the sediments, where the anoxic–oxic interface was near to the bottom. The reservoirs Cubillas, La Bolera, and Francisco Abellán showed this profile type (Figs. S3a, S4a, and S5a). This accumulation of CH₄ in the hypolimnion and above sediments might be related to the high rates of methanogenesis in the sediments and its subsequent diffusion to the water column. Dissolved CH₄ concentration declines at the oxycline level, where the highest rates of CH₄ oxidation usually occur (Oswald et al., 2015, 2016). The CH₄ profiles in this group were similar to the ones found in tropical eutrophic and temperate reservoirs (Naqvi et al., 2018; West et al., 2016). The second profile type presents a small peak of metalimnetic CH₄, concomitant with peaks of dissolved oxygen, chlorophyll $a$, photosynthetic picoeukaryotes, and cyanobacteria (Fig. 3a). In the Negratín reservoir, we found the maximum concentration of CH₄ in the oxic hypolimnion. Unlike several previous works in lakes (Blees et al., 2015; Grossart et al., 2011; Khatun et al., 2019; Murase et al., 2003), we did not find a metalimnetic CH₄ maximum. Khatun et al. (2019) described the existence of a metalimnetic CH₄ maximum in 10 out of 14 lakes. The metalimnetic CH₄ maximum may represent a physically driven CH₄ accumulation due to solubility differences with the temperature at the thermocline, the epilimnetic CH₄ losses by emission, and the lateral inputs from the littoral zone (Donis et al., 2017; Encinas Fernández et al., 2016; Hofmann et al., 2010). The metalimnetic CH₄ maximum can also be determined by biological factors, including the light inhibition of the methane oxidation (Murase and Sugimoto, 2005; Tang et al., 2014) or the distinctive methane production by phytoplankton due to availability of nutrients, light, or precursors at this layer (Khatun et al., 2019). The third profile type included five reservoirs, in which the dissolved CH₄ profile presented a CH₄ accumulation more significant in the epilimnion than in the hypolimnion. The reservoirs Jándula, Bermejales, Rules, El Portillo, and Colomera showed this profile type (Figs. 4a and S6a–9a). These reservoirs had a mean CH₄ concentra-

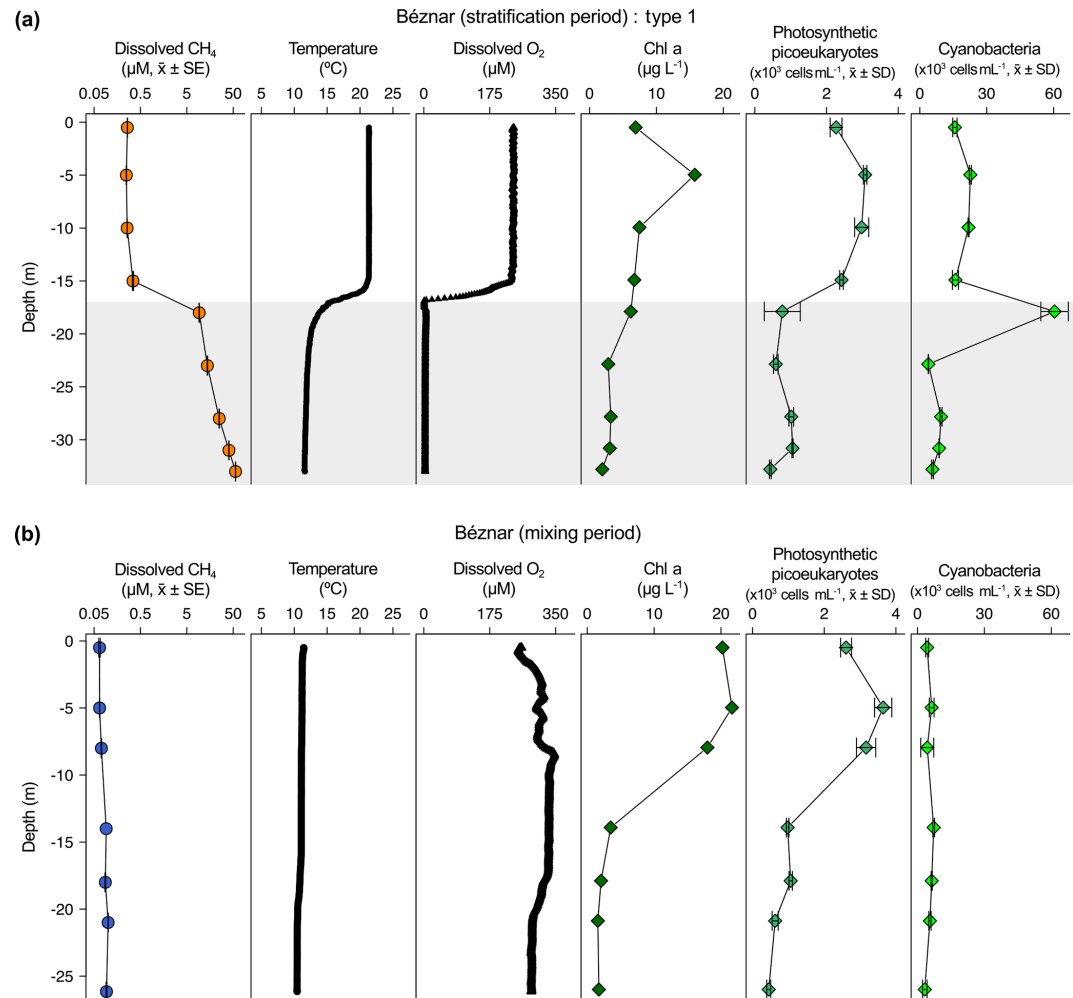

**Figure 2.** CE11 Vertical profiles of physicochemical and biological variables in Béznar reservoir. Dissolved methane concentration (CH$_4$, µM, mean ± standard error), temperature (°C), dissolved oxygen (DO) concentration (µM), chlorophyll $a$ *Chl $a$) concentration (µg L$^{-1}$), abundance of photosynthetic picoeukaryotes ($\times 10^3$ cells mL$^{-1}$, mean ± standard deviation), and abundance of cyanobacteria ($\times 10^3$ cells mL$^{-1}$, mean ± standard deviation) during the stratification period **(a)** and the mixing period **(b)**. The grey area represents the anoxic zone (DO < 7.5 µM). Note the logarithmic scales in the $x$ axis of the dissolved CH$_4$ profiles. The sampling for the stratification period was on 7 October 2016 and 23 February 2017 for the mixing period.

tion in the water column significantly lower than the reservoirs from the first type. Similar profiles have been reported in temperate (Tang et al., 2014) and tropical lakes (Murase et al., 2003).

## 3.2 CH$_4$ sources in the water column

We found two well-differentiated groups of CH$_4$ data sorted by the dissolved oxygen (DO) concentration (Fig. S10), as in previous studies (Tang et al., 2014). The first dataset included the samples with a DO lower than 7.5 µM ($n = 18$, hereafter anoxic samples). These samples belong to the hypolimnion of the studied reservoirs during the stratification period. The second dataset included the samples with DO higher than 7.5 µM ($n = 160$, hereafter oxic samples).

All the samples from the mixing period ($n = 82$) and most of the samples from the stratification period ($n = 78$) belong to this second dataset. We found significant differences ($W = 2632$, $p$ value < 0.001) between the concentration of CH$_4$ in the anoxic samples (median = 15.79, min = 0.35, max = 213.64 µM) and in the oxic samples (median = 0.15, min = 0.02, max = 8.17 µM). Since these two groups of samples are different, we determined their sources and drivers separately (Table S2).

### 3.2.1 CH$_4$ sources in anoxic waters

Archaeal methanogens are obligate anaerobes that decompose the organic matter and produce CH$_4$ in anoxic environments, as freshwater sediments. We analyzed the presence

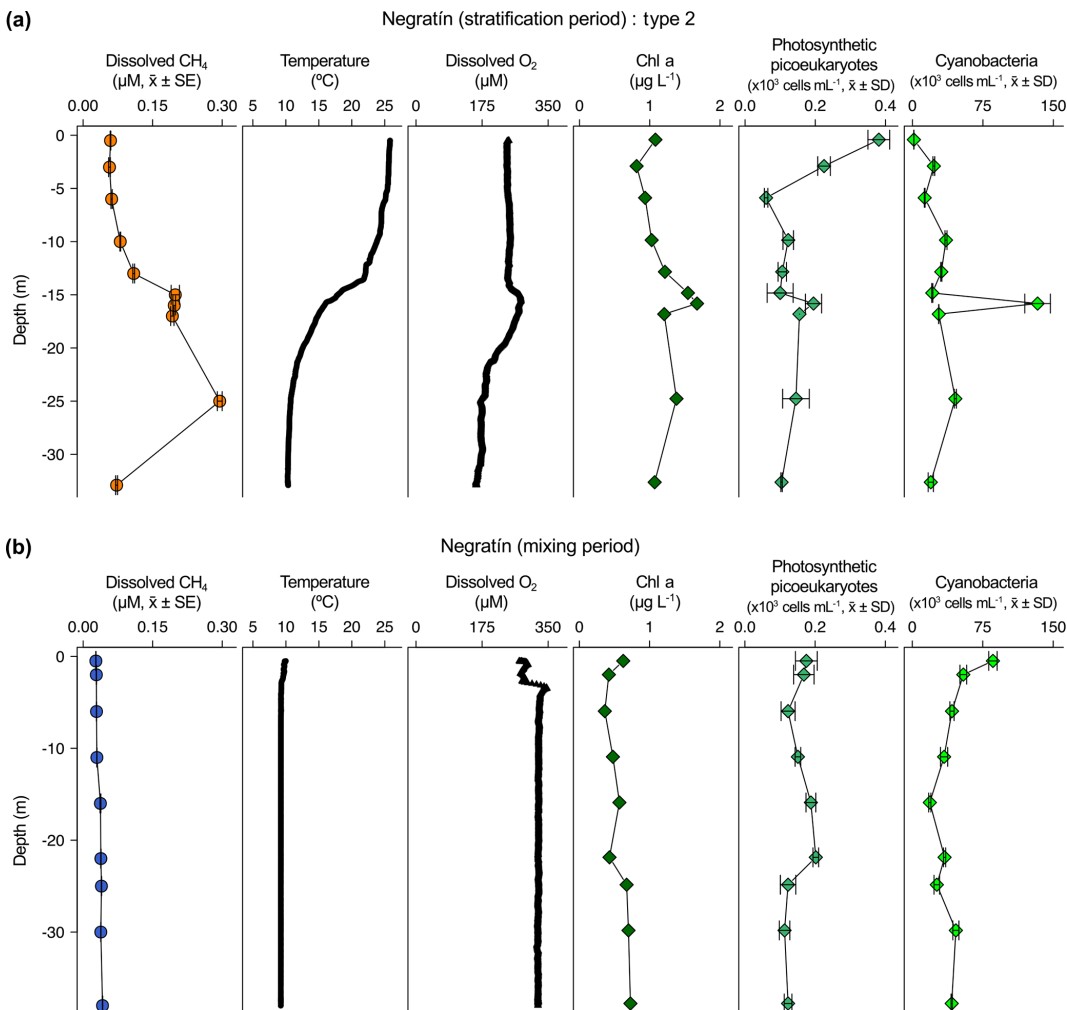

**Figure 3.** Vertical profiles of physicochemical and biological variables in Negratín reservoir. Dissolved methane concentration (CH$_4$, µM, mean ± standard error), temperature (°C), dissolved oxygen (DO) concentration (µM), chlorophyll $a$ (Chl $a$) concentration (µg L$^{-1}$), abundance of photosynthetic picoeukaryotes ($\times 10^3$ cells mL$^{-1}$, mean ± standard deviation), and abundance of cyanobacteria ($\times 10^3$ cells mL$^{-1}$, mean ± standard deviation) during the stratification period **(a)** and the mixing period **(b)**. The sampling for the stratification period was on 27 July 2016 and 16 February 2017 for the mixing period.

of the methanogenic *Archaea* in the anoxic samples of the water column by targeting the gene *mcrA*. From the 77 samples selected for genetic analysis, 12 of them were anoxic. We did not detect the amplification of the *mcrA* gene in the PCR or the qPCR analysis in these 12 samples. Therefore, we assumed that the methanogenic *Archaea* were not present, as free-living microorganisms, in the water column of the anoxic samples. However, they may still be present in micro-anoxic zones in the water column (i.e., in the guts of zooplankton or within exopolymeric particles). Methanogenesis is a microbial process particularly sensitive to temperature (Marotta et al., 2014; Sepulveda-Jauregui et al., 2018; Yvon-Durocher et al., 2014). However, we did not find a significant relationship between the water temperature and the dissolved CH$_4$ concentration in the anoxic samples ($n = 17$,

$p$ value $= 0.66$). The lack of a detection of the *mcrA* gene in the hypolimnetic waters and the absence of a relationship between the dissolved CH$_4$ and water temperature suggest that CH$_4$ production is not happening in the water column of the studied reservoirs. We think that most methanogenic archaea must be present in the sediments, where they produce CH$_4$ that diffuses up to the water column, producing vast accumulations of CH$_4$ in the hypolimnion.

Methanogenesis in the sediments may be affected by organic matter quantity and quality (West et al., 2012). Organic matter quantity is measured as the dissolved organic carbon concentration, whereas the organic matter quality usually is related to their phytoplanktonic versus terrestrial origin. In the studied reservoirs, the dissolved organic carbon concentration did not show a significant relationship with the

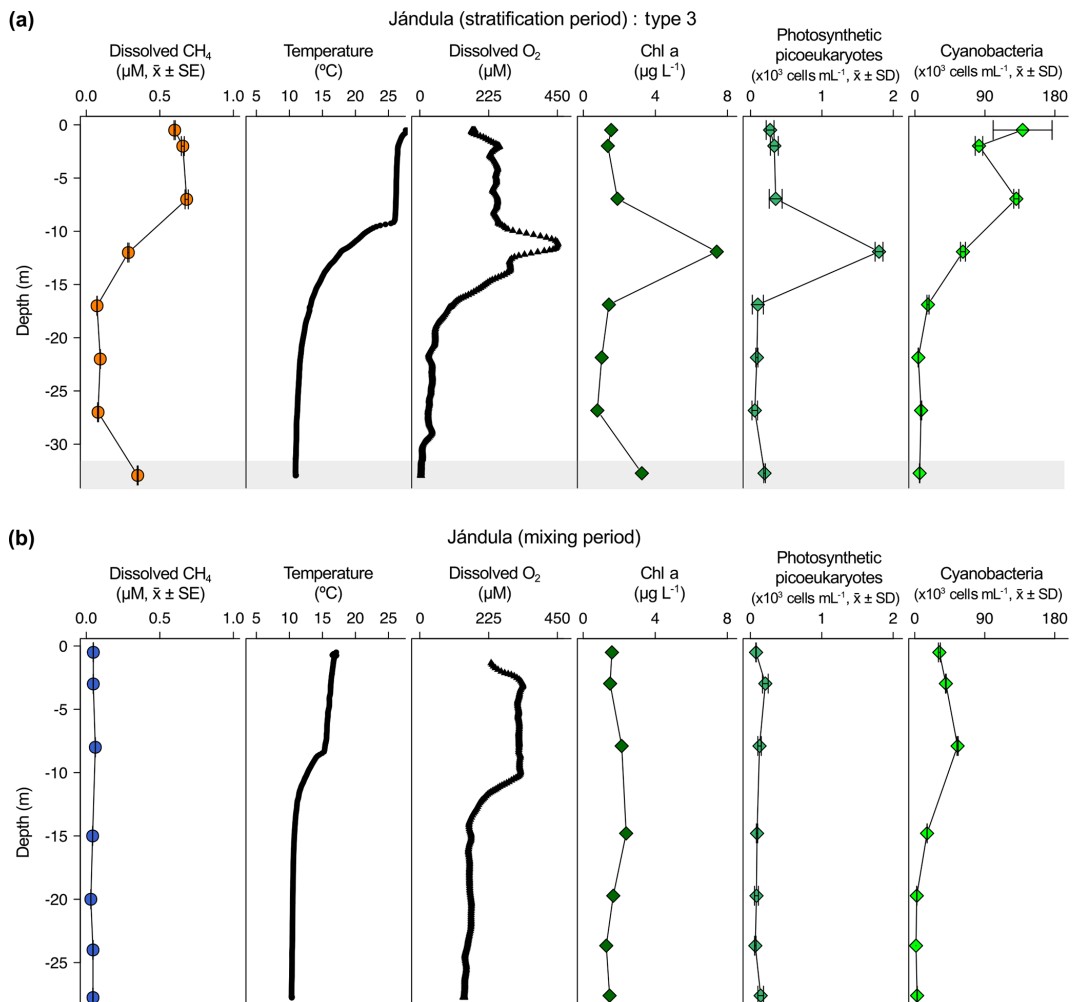

**Figure 4.** Vertical profiles of physicochemical and biological variables in Jándula reservoir. Dissolved methane concentration (CH$_4$, μM, mean ± standard error), temperature (°C), dissolved oxygen (DO) concentration (μM), chlorophyll $a$ (Chl $a$) concentration (μg L$^{-1}$), abundance of photosynthetic picoeukaryotes ($\times 10^3$ cells mL$^{-1}$, mean ± standard deviation), and abundance of cyanobacteria ($\times 10^3$ cells mL$^{-1}$, mean ± standard deviation) during the stratification period **(a)** and the mixing period **(b)**. The grey area represents the anoxic zone (DO $<7.5$ μM). The sampling for the stratification period was on 24 July and 5 April 2017 for the mixing period.

dissolved CH$_4$ concentration ($n = 12$, $p$ value $= 0.10$; Table S2). We examined the importance of the autochthonous organic matter produced by primary producers using the total cumulative Chl $a$ (mg m$^{-2}$). The cumulative Chl $a$ is considered to be a surrogate for the vertical export of the phytoplankton biomass in the whole water column. We found that the CH$_4$ concentrations in anoxic samples were correlated to the cumulative Chl $a$ following a power function (CH$_4 = 3.0 \times 10^{-4}$ cumulative Chl $a^{2.28}$, $n = 17$, adjusted $R^2 = 0.40$, $p$ value $<0.01$; Table S2) (Fig. 5). The autochthonous organic matter appeared to be a better predictor for the concentration of CH$_4$ in anoxic waters than the dissolved organic matter concentration. In the studied reservoirs, the dissolved organic carbon concentration was significantly related to the age of the reservoirs and the forestry coverage in their watersheds (León-Palmero et al., 2019). Therefore, in terms of quality, the total pool of dissolved organic carbon may be more representative of the carbon fraction that is allochthonous, recalcitrant, and more resistant to microbial degradation. In contrast, the autochthonous organic matter may represent a more labile and biodegradable fraction. Previous experimental studies have demonstrated that the addition of algal biomass on sediment cores increases the CH$_4$ production more than the addition of terrestrial organic matter (Schwarz et al., 2008; West et al., 2012, 2015). The stimulation of the methanogenesis rates appears to be related to the lipid content in phytoplankton biomass (West et al., 2015). West et al. (2016) found a significant relationship between the chlorophyll $a$ concentration in the epilimnion and the potential methanogenesis rates from sediment incubations. In this study, we corroborate the importance of the autochthonous-derived organic matter determining the CH$_4$

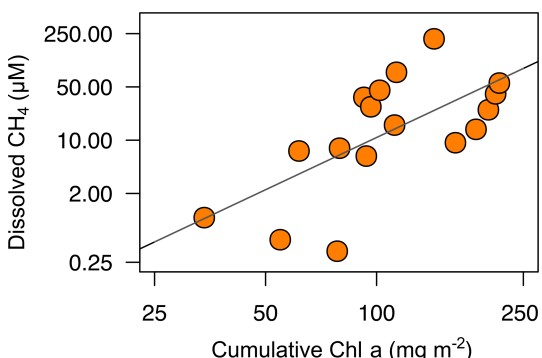

**Figure 5.** Power relationship between the depth-cumulative chlorophyll *a* concentration and the concentration of dissolved CH$_4$ in the anoxic waters during the stratification period (CH$_4$, μM $= 3.0 \times 10^{-4}$ cumulative Chl $a^{2.28}$, $n = 17$, adjusted $R^2 = 0.40$). Note that both axes are at the logarithmic scale. More statistical details can be found in Table S2.

concentrations in anoxic waters. Since we did not detect the existence of the *mcrA* gene in the water column, we considered that the production of methane by methanogenic *Archaea* occurred primarily in the sediments and was affected by the sedimentation of organic matter derived from phytoplankton.

### 3.2.2   CH$_4$ sources in oxic waters

In this study, the concentration of dissolved CH$_4$ ranged from 0.02 to 8.18 μM, and all the samples of the oxic waters were supersaturated, with values always above 800 % and ranging more than 2 orders of magnitude (Table S1). To determine the origin of this CH$_4$ supersaturation, we examined the following potential sources: (1) the vertical and lateral CH$_4$ transport from deep layers and littoral zones, (2) the in situ CH$_4$ production by methanogenic *Archaea* potentially tolerant to oxygen or by the methylphosphonate degradation under severe P limitation, and (3) the in situ CH$_4$ production by processes associated to the phytoplanktonic community.

**Vertical and lateral CH$_4$ transport from anoxic sediments to oxic waters**

Several previous works have pointed out that CH$_4$ supersaturation in oxic waters can be explained by the vertical transport from the bottom sediments and the lateral inputs from the littoral zones that are in contact with shallow sediments where methanogenesis occurs (Bastviken et al., 2004; Encinas Fernández et al., 2016; Michmerhuizen et al., 1996). To test the importance of the lateral and vertical transport explaining the concentration of CH$_4$ in the oxic waters of the studied reservoirs, we used two morphometric parameters: the mean depth (m) as a proxy for the vertical transport and the shallowness index as a proxy for the lateral transport. The dissolved CH$_4$ concentration was an exponential decay

function of the reservoir mean depth (Fig. 6a) both during the stratification period (CH$_4 = 4.0 \times 10^{-2} e^{(50.0/\text{mean depth})}$, adjusted $R^2 = 0.95$) and during the mixing period (CH$_4 = 3.7 \times 10^{-2} e^{(22.9/\text{mean depth})}$, adjusted $R^2 = 0.54$) (Fig. 6a). We observed that in reservoirs with a mean depth shallower than 16 m, the dissolved CH$_4$ concentration increased exponentially (Fig. 6a). Several studies have proposed that the vertical transport of CH$_4$ from bottom sediments explains the supersaturation in surface waters (Rudd and Hamilton, 1978; Michmerhuizen et al., 1996; Murase et al., 2003; Bastviken et al., 2004). However, the vertical diffusion rates of dissolved gases across the thermocline are too low in deep and thermally stratified systems, and no movements of methane upwards from the hypolimnion have been detected (Rudd and Hamilton, 1978). However, in shallow reservoirs, the hydrostatic pressure might be reduced, promoting CH$_4$ diffusion from the anoxic layers.

The shallowness index increases in elongated and dendritic reservoirs, with a greater impact of the littoral zone, and decreases in near-circular reservoirs, with low shoreline length per surface. However, we did not find a significant relationship between the shallowness index and the dissolved CH$_4$ concentration (Fig. 6b). One explanation for the absence of this relationship could be the relatively large size of the reservoirs. Although the reservoir size covered more than 1 order of magnitude (Table 1), all reservoirs have a size larger than 1 km$^2$. Previous studies have shown that CH$_4$ lateral diffusion may be an important process in areas near to the littoral zone and small lakes. Hofmann et al. (2010) found higher concentrations in the shallow littoral zones than in the open waters. DelSontro et al. (2018) predicted that lateral inputs from littoral zones to pelagic waters are more critical in small and round lakes than in large and elongated lakes. Nevertheless, the differences between the observations and predictions from the model suggested that these lateral inputs may not be enough to explain CH$_4$ concentration in open waters, where in situ production may prevail over lateral transport (DelSontro et al., 2018).

**In situ CH$_4$ production by methanogenic *Archaea* or methylphosphonate degradation**

The ubiquitous CH$_4$ supersaturation found in oxic waters appears not to be fully explained by the vertical and lateral transport, underlining that there is an in situ production of CH$_4$, as proposed by Bogard et al. (2014), DelSontro et al. (2018), and Grossart et al. (2011). We studied the presence of the methanogenic *Archaea* in the oxic samples by targeting the gene *mcrA*, but we were unable to detect this gene (Fig. S11). This result indicates that methanogenic *Archaea* were not present, at least as free-living microorganisms, in a significant number in the water column of the oxic samples. The classical methanogens (i.e., *Archaea* with the *mcrA* gene) are obligate anaerobes without the capacity to survive and produce CH$_4$ under aerobic conditions (Chistoserdova et

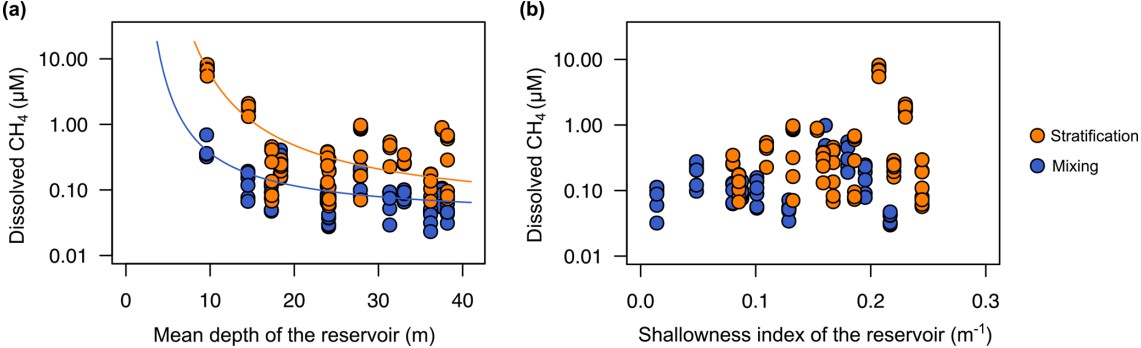

**Figure 6.** Reservoir morphometry and the dissolved CH$_4$ concentration in the oxic zone. **(a)** Exponential decay relationships of the dissolved CH$_4$ concentration and the mean depth (m) during the stratification period (CH$_4$ = $4.0 \times 10^{-2} e^{(50.0/\text{mean depth})}$, $n = 78$, adjusted $R^2 = 0.95$) and the mixing period (CH$_4$ = $3.7 \times 10^{-2} e^{(22.9/\text{mean depth})}$, $n = 82$, adjusted $R^2 = 0.54$). **(b)** Scatterplot of dissolved CH$_4$ concentration and the reservoir shallowness index during the stratification period ($p$ value = 0.134) and the mixing period ($n = 0.114$). More statistical details can be found in Table S2.

al., 1998). Previous studies by Angel et al. (2011) and Angel TS7 et al. (2017) showed that methanogens might tolerate oxygen exposure in soils, and Grossart et al. (2011) detected potential methanogenic *Archaea* attached to photoautotrophs in oxic lake waters. Unfortunately, we did not test their occurrence in large particles, phytoplankton, or zooplankton guts, although some authors have detected them in these microsites' particles (de Angelis and Lee, 1994; Karl and Tilbrook, 1994).

We also considered the possibility of methylphosphonate degradation as an in situ CH$_4$ source. This metabolic pathway appears in the bacterioplankton under chronic starvation for phosphorus (Karl et al., 2008). Several pieces of evidence have shown that marine bacterioplankton can degrade the MPn's and produce CH$_4$ through the C–P lyase activity in typically phosphorus-starved environments, like the ocean gyres (Beversdorf et al., 2010; Carini et al., 2014; Repeta et al., 2016; Teikari et al., 2018; del Valle and Karl, 2014). Freshwater bacteria can also degrade the MPn's and produce CH$_4$, as has been demonstrated in Lake Matano (Yao et al., 2016a, b). Lake Matano is an ultra-oligotrophic lake with a severe P deficiency (below 0.050 µmol P L$^{-1}$) due to the permanent stratification, iron content, and extremely low nutrient inputs (Crowe et al., 2008; Sabo et al., 2008). The ratio of dissolved inorganic nitrogen (DIN) to total phosphorus (TP) (µmol N : µmol P) is widely used to evaluate P limitation (Morris and Lewis, 1988). DIN : TP ratios greater than 4 are indicative of phosphorus limitation (Axler et al., 1994). In the studied reservoirs, the TP concentration ranged from 0.13 to 1.85 µmol P L$^{-1}$ during the stratification period and from 0.10 to 2.17 µmol P L$^{-1}$ during the mixing period. The DIN : TP ratio ranged from 15 to 985 during the stratification period and from 28 to 690 during the mixing period. The more severe the P limitation conditions are, the higher the CH$_4$ production by methylphosphonates degradation is. However, we did not find a significant relationship be-

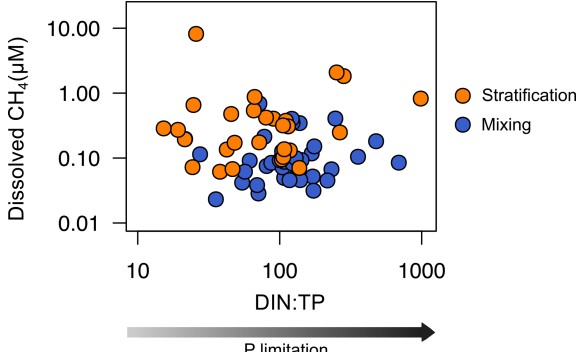

**Figure 7.** Phosphorus limitation and the dissolved CH$_4$ concentration in the oxic waters. Scatterplot of dissolved CH$_4$ concentration and the ratio between dissolved inorganic nitrogen (DIN) and the total phosphorus (TP) (µmol N : µmol P). Note the logarithmic scale in both axes.

tween the DIN : TP ratio and the CH$_4$ concentration (Fig. 7). We also analyzed the presence and abundance of the gene *phnJ*, which encodes the enzyme complex C–P lyase that hydrolyzes the MPn's and changes in response to phosphate availability. We did not detect the *phnJ* gene in the PCR or the qPCR analysis in any of the study samples (Fig. S12). These results indicate that the MPn degradation was not a quantitatively relevant source of CH$_4$ in the oxic waters of the studied reservoirs. Our results are in concordance with Grossart et al. (2011), who did not detect CH$_4$ production by adding inorganic phosphate or methylphosphonates to lake samples in laboratory experiments. Although we used different methodologies, both studies may indicate that MPn degradation is only an important source of CH$_4$ in ultra-oligotrophic systems, as in Lake Matano or ocean gyres.

## In situ $CH_4$ production coupled to photosynthetic organisms

In the studied reservoirs, we analyzed the relationship between photosynthetic organisms and the dissolved $CH_4$ concentration using the GPP ($g\,O_2\,m^{-3}\,d^{-1}$), NEP ($g\,O_2\,m^{-3}\,d^{-1}$), the concentration of Chl $a$ ($\mu g\,L^{-1}$), and the abundance of photosynthetic picoeukaryotes (PPEs; cells $mL^{-1}$) and cyanobacteria (CYA; cells $mL^{-1}$). We determined GPP and NEP just once per reservoir during the stratification period (i.e., $n = 12$).

The PPEs are essential components of the marine and freshwater phytoplankton, and they are eukaryotes with a size of 3.0 µm or less. In the freshwater, the PPEs include species from different phyla, like unicellular *Chlorophyta* (green algae) and *Haptophyta*. Using optical microscopy, we determined the main groups of photosynthetic picoeukaryotes in the studied reservoirs. PPEs were non-colonial green algae from the order *Chlorococcales* (class *Chlorophyceae*, phylum *Chlorophyta*) CE12 and the genus *Chrysochromulina* spp. (class *Coccolithophyceae*, phylum *Haptophyta*). The cyanobacteria detected were mainly phycoerythrin-rich picocyanobacteria, although we also detected phycocyanin-rich picocyanobacteria in one reservoir (Béznar). We show the vertical profiles of the Chl $a$ concentration and the abundance of PPEs and CYA profiles of each reservoir in Figs. 2–4 and S1–S9. We also report the minimum, the quartiles, and the maximum values for the Chl $a$ concentration and the abundance of PPEs and CYA during the stratification and the mixing periods in Table S2. The abundance of cyanobacteria ranged from $1.51 \times 10^3$ to $2.04 \times 10^5$ cells $mL^{-1}$ and was more than 1 order of magnitude higher than the abundance of PPEs that ranged from 32 to $7.45 \times 10^3$ cells $mL^{-1}$.

We found that the relationship between the gross primary production and the dissolved $CH_4$ concentration was only marginally significant ($p$ value $= 0.077$, $n = 12$) and not significant with the net ecosystem production (Table 3). The Chl $a$ concentration showed a significant relationship with the GPP ($p$ value $<0.01$, $n = 12$, adjusted $R^2 = 0.55$), but the abundance of cyanobacteria or the abundance of the photosynthetic picoeukaryotes did not show a significant relationship with the GPP ($p$ value $= 0.911$, $n = 12$, and $p$ value $= 0.203$, $n = 12$, respectively). We found significant power relationships between the Chl $a$ concentration, the abundance of photosynthetic picoeukaryotes, and the abundance of cyanobacteria with the concentration of dissolved $CH_4$ during the stratification period (Fig. 8a, b, and c, respectively, and Table 3). During the mixing period, the only significant predictor of the dissolved $CH_4$ concentration was the abundance of photosynthetic picoeukaryotes (Fig. 8b). The variance was explained, and the slope of the relationship (i.e., the exponent in the power relationship) between the dissolved $CH_4$ and the abundance of photosynthetic picoeukaryotes was higher during the stratification than during the mixing (Table 3). By comparing the stratification slopes,

the effect per cell of PPEs on $CH_4$ concentration was slightly higher than the impact of cyanobacteria (Table 3). These results agree with previous studies that showed a closed link between the $CH_4$ concentration and the photosynthetic organisms, primary production, or chlorophyll $a$ concentration (Bogard et al., 2014; Grossart et al., 2011; Schmidt and Conrad, 1993; Tang et al., 2014). In this study, we show that the PPE abundance was a better predictor of the $CH_4$ concentration than the abundance of cyanobacteria. In the studied reservoirs, the PPE group included members from green algae and *Haptophyta*, which are regular components of the marine plankton. Therefore, these results may also be relevant for marine waters. Cyanobacteria have received more attention as potential producers of $CH_4$ in oxic conditions than photosynthetic picoeukaryotes (Berg et al., 2014; Bižić et al., 2020; Teikari et al., 2018). Klintzsch et al. (2019) demonstrated that widespread marine and freshwater haptophytes like *Emiliania huxleyi*, *Phaeocystis globosa*, and *Chrysochromulina* sp. produce $CH_4$ under oxic conditions. They also observed that the cell abundances were significantly related to the amount of $CH_4$ produced. Interestingly, *Chrysochromulina* was one of the genera of PPEs that we detected in the studied reservoirs. Grossart et al. (2011) also found $CH_4$ production in laboratory cultures of cyanobacteria and green algae.

Overall, these results indicate a clear association between the $CH_4$ production and the photosynthetic organisms from both *Eukarya* (picoeukaryotes) and *Bacteria* (cyanobacteria) domains. The pathways involved in the $CH_4$ production may be related to the central photosynthetic metabolism or the release of methylated by-products, different from methylphosphonates during the photosynthesis. Previous studies demonstrated the $CH_4$ production in laboratory cultures using $^{13}$C-labeled bicarbonate in haptophytes (Klintzsch et al., 2019; Lenhart et al., 2016); in marine, freshwater, and terrestrial cyanobacteria (Bižić et al., 2020); and in major groups of phytoplankton (Hartmann et al., 2020). In these studies, the photosynthetic organisms uptake bicarbonate in the reductive pentose phosphate cycle (Calvin–Benson cycle) (Berg, 2011; Burns and Beardall, 1987). Therefore, $CH_4$ production may be a common pathway in the central metabolism of photosynthesis of all the cyanobacteria and algae in freshwater and marine environments.

On the other hand, the production of $CH_4$ can also be related to the production of methylated compounds during photosynthesis. Lenhart et al. (2016) and Klintzsch et al. (2019) also detected the $CH_4$ production in cultures from the sulfur-bound methyl group of the methionine and methyl thioethers. Common substances like methionine can act as a methyl-group donor during the $CH_4$ production in plants and fungi (Lenhart et al., 2012, 2015). Besides, algae use part of the methionine for the synthesis of dimethylsulfoniopropionate (DMSP), an abundant osmolyte, the precursor of dimethyl sulfide (DMS), and dimethyl sulfoxide (DMSO). These methylated substances produce methane during their

**Table 3.** Equations for the relationships between the phytoplanktonic variables and the dissolved CH$_4$ concentration in the oxic waters. n.m. means not measured.

| Driver | Period | $n$ | Equation | Adjusted $R^2$ | $p$ value |
|---|---|---|---|---|---|
| Chl $a$ concentration ($\mu$g L$^{-1}$) | Stratification + mixing | 160 | CH$_4$ ($\mu$M) $= 0.12$ Chl $a^{0.44}$ | 0.11 | $<0.001$ |
| | Stratification | 78 | CH$_4$ ($\mu$M) $= 0.14$ Chl $a^{0.97}$ | 0.40 | $<0.001$ |
| | Mixing | 82 | Not significantly related | | 0.469 |
| Gross primary production (GPP; g O$_2$ m$^{-3}$ d$^{-1}$) | Stratification | 12 | Marginally significant | | 0.077 |
| | Mixing | n.m. | | | |
| Net ecosystem production (NEP; g O$_2$ m$^{-3}$ d$^{-1}$) | Stratification | 12 | Not significantly related | | 0.536 |
| | Mixing | n.m. | | | |
| Photosynthetic picoeukaryotes' (PPEs') abundance (cells mL$^{-1}$) | Stratification + mixing | 160 | CH$_4$ ($\mu$M) $= 2.0 \times 10^{-2}$ PPEs$^{0.35}$ | 0.19 | $<0.001$ |
| | Stratification | 78 | CH$_4$ ($\mu$M) $= 7.2 \times 10^{-3}$ PPEs$^{0.65}$ | 0.57 | $<0.001$ |
| | Mixing | 82 | CH$_4$ ($\mu$M) $= 3.2 \times 10^{-2}$ PPEs$^{0.16}$ | 0.12 | $<0.001$ |
| Cyanobacteria (CYA) abundance (cells mL$^{-1}$) | Stratification + mixing | 160 | CH$_4$ $\mu$M $= 9.9 \times 10^{-4}$ CYA$^{0.53}$ | 0.19 | $<0.001$ |
| | Stratification | 78 | CH$_4$ $\mu$M $= 1.7 \times 10^{-3}$ CYA$^{0.53}$ | 0.17 | $<0.001$ |
| | Mixing | 82 | Not significantly related | | 0.666 |

degradation (Damm et al., 2008, 2010, 2015; Zindler et al., 2013). Bižić-Ionescu et al. (2018) also suggested that CH$_4$ could be produced from methylated amines under oxic conditions. These substances, together with other organosulfur compounds, can also produce CH$_4$ abiotically (Althoff et al., 2014; Bižić-Ionescu et al., 2018). The production of DMSP, DMS, and other methylated substances like isoprene has been extensively studied in marine phytoplankton, showing that taxa as photosynthetic picoeukaryotes and the cyanobacteria are relevant sources (Shaw et al., 2003; Yoch, 2002). Recent studies have also reported that freshwater algae and cyanobacteria also produced DMS and isoprene (Steinke et al., 2018). Further studies are needed to quantify the potential role of all these methylated by-products as potential CH$_4$ sources quantitatively relevant in freshwater.

### 3.2.3 Modeling the CH$_4$ production in oxic waters

The explanation of the CH$_4$ supersaturation in oxic waters in relatively large systems relies on the interaction of several processes as the transport from anoxic environments and the biological activity (DelSontro et al., 2018). In this study, we found that vertical transport (mean depth as surrogate), water temperature, and the abundance of photosynthetic picoeukaryotes and cyanobacteria had a significant effect on the dissolved CH$_4$ concentration. We combined these explanatory variables with significant effects using GAMs. The GAM for the stratification period ($n = 78$) had a fit deviance of 82.7 % and an explained variance (adjusted $R^2$) of 81.4 % (Table S3). The explanatory variables, in decreasing order, were as follows: the photosynthetic picoeukaryotes' abundance (log$_{10}$ PPEs), the reservoir mean depth, the cyanobacteria abundance (log$_{10}$ CYA), and the water temperature (Fig. 9a). The function obtained was as follows: log$_{10}$ CH$_4$ $= -4.05 + 3.4 \times 10^{-1}$ log$_{10}$ PPEs $+ e^{(6.7/\text{mean depth})} + 1.7 \times 10^{-1}$ log$_{10}$ CYA $+ 2.7 \times 10^{-2}$ Temperature. The abundance of PPEs was the variable explaining most of the variance of dissolved CH$_4$ concentration (log$_{10}$ CH$_4$) during the stratification period, with an effect higher than the cyanobacteria abundance. Figure 9b–e shows the partial responses of each explanatory variable.

The GAM for the mixing period ($n = 82$) only included two explanatory variables: the reservoir mean depth and the abundance of the photosynthetic picoeukaryotes. The reservoir mean depth was the variable explaining most of the variance of the dissolved CH$_4$ concentration (log$_{10}$ CH$_4$) during the mixing period, closely followed by the abundance of PPEs (Fig. 10a). We observed that the function of the effect of the mean depth on the CH$_4$ concentration changed between the two periods (Figs. 9c and 10b). The function was more linear during the mixing period than during the stratification period, likely because the mixed water column enabled the more uniform distribution of the CH$_4$ produced in the sediment, while the thermocline acted as a barrier to the diffusion during the stratification period. The model function for the mixing period was log$_{10}$ CH$_4$ $= -2.07 + 1.5 e^{(-0.04 \text{ mean depth})} + 1.8 \times 10^{-1}$ log$_{10}$ PPEs, with a fit deviance of 53.9 % and an explained variance (adjusted $R^2$) of 52.1 % (Table S3). In Fig. 10b and c, we show the partial response plots for these two variables. The re-

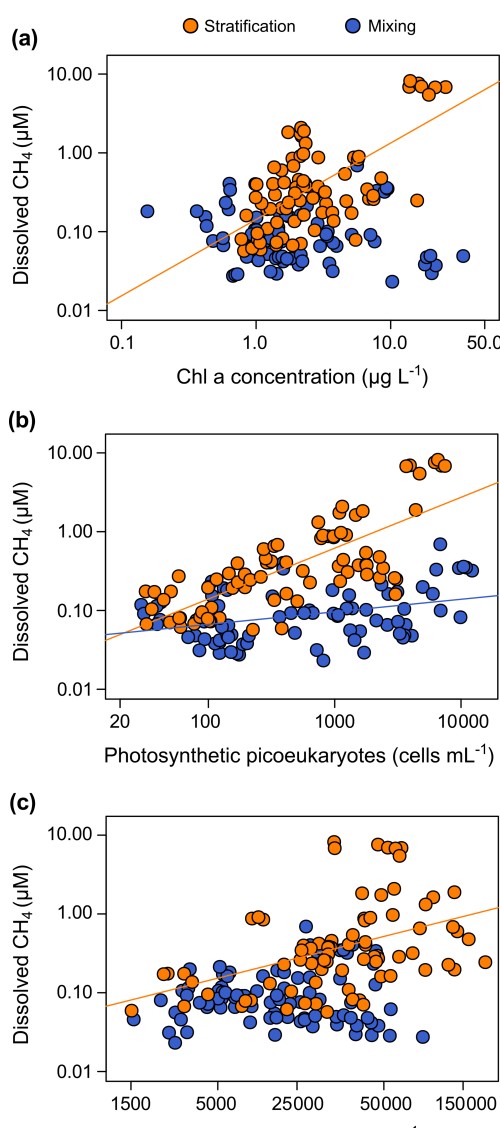

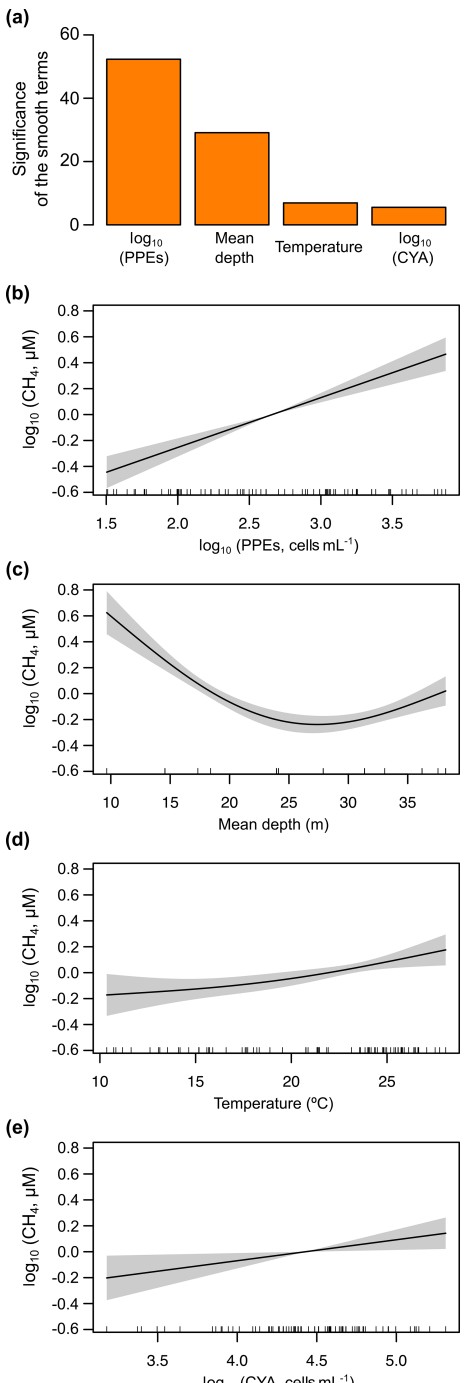

**Figure 8.** Phytoplanktonic variable coupled with the dissolved $CH_4$ concentration in the oxic waters. **(a)** The dissolved $CH_4$ concentration was significantly related to the chlorophyll $a$ concentration during the stratification period ($p$ value $<0.001$), but they were not related during the mixing period ($p$ value $= 0.469$). The relationship during the stratification period was a power function ($CH_4$, $\mu M = 0.14$ Chl $a^{0.97}$, $n = 78$, adjusted $R^2 = 0.40$). **(b)** Relationships between dissolved $CH_4$ concentration and the abundance of photosynthetic picoeukaryotes (PPEs) during the stratification period ($CH_4$, $\mu M = 7.2 \times 10^{-3}$ PPEs$^{0.65}$, $n = 78$, adjusted $R^2 = 0.55$, $p$ value $<0.001$) and the mixing period ($CH_4$, $\mu M = 3.2 \times 10^{-2}$ PPEs$^{0.16}$, $n = 82$, adjusted $R^2 = 0.12$, $p$ value $<0.001$). **(c)** Relationship between dissolved $CH_4$ concentration and the cyanobacteria abundance (CYA; cells mL$^{-1}$). A power function described the relationship between the dissolved $CH_4$ and the CYA during the stratification period ($CH_4$, $\mu M = 1.7 \times 10^{-3}$ CYA$^{0.53}$, $n = 78$, adjusted $R^2 = 0.17$, $p$ value $<0.001$). The relationship was not significant during the mixing period ($p$ value $= 0.666$).

**Figure 9.** Results of the generalized additive model (GAM) fitted for the concentration of dissolved $CH_4$ in the oxic waters during the stratification period. **(a)** Bar plot showing the significance of the smooth terms from the fitted GAM ($F$ values). **(b–e)** Partial response plots from the fitted GAM, showing the additive effects of the covariates on the dissolved $CH_4$ concentration: the photosynthetic picoeukaryotes' abundance ($\log_{10}$ PPEs) **(b)**, the mean depth **(c)**, the cyanobacteria abundance ($\log_{10}$ CYA) **(d)**, and water temperature **(e)**. In partial response plots, the lines are the smoothing functions, and the shaded areas represent 95 % pointwise confidence intervals. Rugs on $x$ axis indicate the distribution of the data. More details are provided in Table S3.

sults show that the abundance of photosynthetic picoeukaryotes can be key for explaining the dissolved CH$_4$ concentration in oxic waters, even though they have received less attention than cyanobacteria in previous studies (Berg et al., 2014; Bižić et al., 2020; Teikari et al., 2018). Finally, we have also included a simple model to explain the dissolved CH$_4$ concentration (log$_{10}$ CH$_4$) using the data of both periods ($n = 160$) and including widely used variables like the water temperature (°C), mean depth (m), and Chl $a$ concentration (μg L$^{-1}$) for future comparisons. The function of this model is $\log_{10}\text{CH}_4\,(\mu\text{M}) = -2.02 + 0.05\,\text{Temperature} + e^{(7.73/\text{mean depth})} - e^{(-0.05\log_{10}(\text{Chl } a))}$. This GAM had a fit deviance of 69.3 % and an explained variance (adjusted $R^2$) of 68 % (Table S3).

Overall, during the stratification period, the in situ CH$_4$ production was coupled to the abundance of photosynthetic picoeukaryotes in oxic waters (Fig. 9a) and mean depths. This CH$_4$ source, due to photosynthetic picoeukaryotes, can be crucial in large, deep lakes and reservoirs and the open ocean, since the impact of the CH$_4$ transport from sediments (i.e., mean depth) decreases with increasing depths. In deeper reservoirs, the thermal stratification during the summer that produced the vertical diffusion rates of CH$_4$ from sediments is limited. Rudd and Hamilton (1978) did not detect any movement of CH$_4$ upwards from the hypolimnion during the stratification. Previous studies have suggested that the CH$_4$ produced in the oxic water column is the primary source of CH$_4$ in large and deep lakes (Bogard et al., 2014; DelSontro et al., 2018; Donis et al., 2017; Günthel et al., 2019). Günthel et al. (2019) showed that large lakes have a lower sediment area in comparison to the volume of the surface mixed layer than small lakes and that this fact determines the higher contribution of the oxic methane production to surface emission in large (>1 km$^2$) lakes than in small ones. The photosynthetic picoeukaryotes identified in the studied reservoirs are considered indicators of eutrophic conditions, and they are bloom-forming genera (i.e., *Chlorococcales* and *Chrysochromulina* spp.) (Edvardsen and Paasche, 1998; Reynolds, 1984; Willén, 1987). Global future estimations suggest a rise in eutrophication and algal bloom over the next century due to climate change and the growing human population (Beaulieu et al., 2019). In that situation, photosynthetic picoeukaryotes like *Chlorococcales* and *Chrysochromulina* spp., and cyanobacteria, would lead to an increment in CH$_4$ production and emissions. Further studies are needed to understand the role of the photosynthetic picoeukaryotes in the production of CH$_4$ in oxic waters better and to quantify their influence in the methane supersaturation and CH$_4$ fluxes from inland and oceanic waters.

## 4   Conclusions

The dissolved CH$_4$ concentration in the studied reservoirs showed a considerable variability (i.e., up to 4 orders of

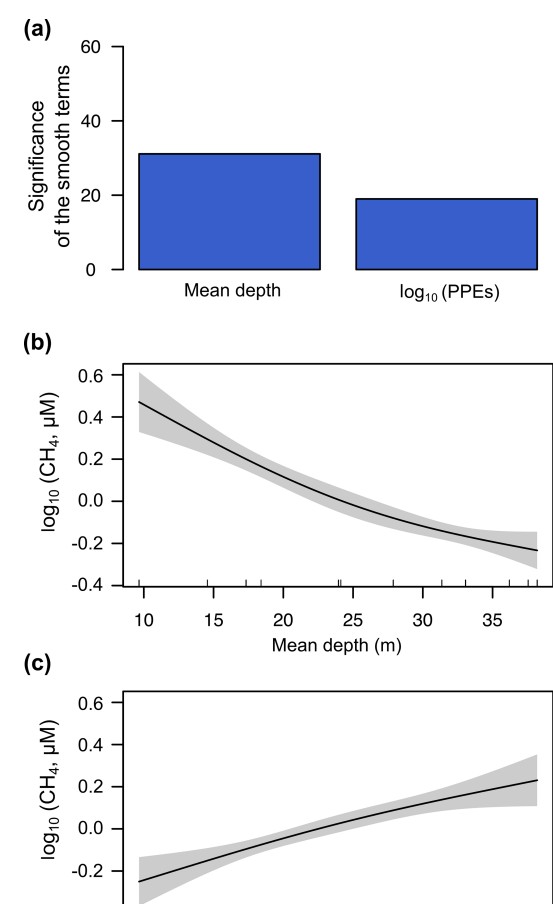

**Figure 10.** Results of the generalized additive model (GAM) fitted for the concentrations of CH$_4$ in the oxic waters during the mixing period. **(a)** Bar plot showing the significance of the smooth terms from the fitted GAM ($F$ values). Panels **(b)** and **(c)** show partial response plots from the fitted GAM, showing the additive effects of the covariates on the dissolved CH$_4$ concentration: the mean depth **(b)** and the abundance of photosynthetic picoeukaryotes (log$_{10}$ PPEs) **(c)**. In partial response plots, the lines are the smoothing functions, and the shaded areas represent 95 % pointwise confidence intervals. Rugs on $x$ axis indicate the distribution of the data. More details are provided in Table S3.

magnitude) and presented a clear seasonality. Surface waters were always supersaturated in CH$_4$. The concentration of CH$_4$ was closely linked to the photosynthetic organisms. In the anoxic waters, the depth-cumulative chlorophyll $a$ concentration, a proxy for the phytoplanktonic biomass exported to sediments, determined the CH$_4$ concentration. In the oxic waters, we considered different potential CH$_4$ sources, including the vertical and lateral transport of CH$_4$ from anoxic zones and in situ production. The mean depth of the reservoirs, as a surrogate of the CH$_4$ transport from sediment to the oxic waters, contributed in shallow systems. We did not

detect methanogenic *Archaea* or methylphosphonates degradation target genes (i.e., *mcrA* and *phnJ* genes, respectively), which suggests that these pathways are not responsible for the in situ production of $CH_4$ in the oxic waters of the studied reservoirs. We found that dissolved $CH_4$ was coupled to the abundance of photosynthetic picoeukaryotes (PPEs) during both periods and to chlorophyll *a* concentration and the abundance of and cyanobacteria during the stratification period. These PPEs were non-colonial green algae from the order *Chlorococcales* (class *Chlorophyceae*, phylum *Chlorophyta*) and the genus *Chrysochromulina* spp. (class *Coccolithophyceae*, phylum TS8 *Haptophyta*). Finally, we combined all the explanatory variables with significant effects and determined their relative contribution to the $CH_4$ concentration using generalized additive models (GAMs). The abundance of PPEs was the variable explaining most of the variance of dissolved $CH_4$ concentration during the stratification period, with an effect higher than the cyanobacteria abundance. During the mixing period, the reservoir mean depth and the abundance of the PPEs were the only drivers for $CH_4$ concentration. Our findings show that the abundance of PPEs can be relevant for explaining the dissolved $CH_4$ concentration in oxic waters of large lakes and reservoirs.

*Data availability.* Additional figures and tables can be found in the Supplement. The dataset associated with this paper will be available at PANGAEA: CE13 Dissolved concentrations of $CH_4$, nutrients, and biological parameters in the water column of twelve Mediterranean reservoirs in Southern Spain (https://doi.org/10.1594/PANGAEA.912535 TS9) and Primary production of twelve Mediterranean reservoirs in Southern Spain (https://doi.org/10.1594/PANGAEA.912555 TS10).

*Supplement.* The supplement related to this article is available online at: https://doi.org/10.5194/bg-17-1-2020-supplement.

*Author contributions.* ELP, RMB, and IR contributed equally to this work. RMB and IR designed the study and obtained the funds. ELP, RMB, and IR contributed to data acquisition during the reservoir samplings. ELP processed most of the chemical and biological samples. ACR performed the flow cytometry and part of the molecular analysis, and AS collaborated with the dissolved $CH_4$ analysis using gas chromatography. ELP, RMB, and IR analyzed the data and discussed the results. ELP wrote the first draft of the paper, which was complemented by significant contributions from RMB and IR.

*Competing interests.* The authors declare that they have no conflict of interest.

*Acknowledgements.* This research was funded by the project HERA (grant no. CGL2014-52362-R) to Isabel Reche and Rafael Morales-Baquero of the Spanish Ministry of Econ-omy and Competitiveness, the Modelling Nature Scientific Unit (grant no. UCE.PP2017.03), and the Consejería de Economía, Conocimiento, Empresas, y Universidad from Andalucia and the European Regional Development Fund (ERDF), reference no. SOMM17/6109/UGR, to Isabel Reche. Elizabeth León-Palmero and Ana Sierra were supported by PhD fellowships from the Ministry of Education, Culture and Sport (grant nos. FPU014/02917 and FPU2014-04048, respectively). Alba Contreras-Ruiz was supported by the Youth Employment Initiative (YEI) from the Junta de Andalucía and financed by the European Commission (reference no. 6017). We especially thank Eulogio Corral for helping in the field, Jesús Forja and Teodora Ortega for helping with gas chromatography analysis at the University of Cádiz, and David Fernández Moreno from the Department of Botany at the University of Granada for the taxonomical identification of the phytoplankton community. We thank the Hydrological Confederations of Guadalquivir and Sur for facilitating the reservoir sampling.

*Financial support.* This research has been supported by the Spanish Ministry of Economy and Competitiveness (grant no. CGL2014-52362-R) and the University of Granada – Unidades de Excelencia CE14 (grant no. UCE.PP2017.03). TS11

*Review statement.* This paper was edited by Carolin Löscher and reviewed by three anonymous referees.

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

## Remarks from the language copy-editor

CE1   Please check and confirm the affiliation edits. Per house standards, departments were listed before their respective institutions. If two departments are listed, the affiliation should be split in two. Thanks!

CE2   Please note that this manuscript has undergone copy-editing according to the standards of American English.

CE3   Please note that per house standards on taxonomic names, divisions higher than genus – phylum, class, order, and family – are capitalized but not italicized. Genus and species names are capitalized and italicized. Please check the text to see if adjustment is necessary. Thanks!

CE4   Please see above comment.

CE5   Please note that as a house standard, we add "s" to plural forms of acronyms. If there are lowercase letters in the abbreviation, "'s" is added for the plural form. Please check the changes made throughout.

CE6   Should this read "in duplicate" and "in triplicate" throughout?

CE7   Does "M" stand for "mole"? If so, this should read "mol"(also in Table 2).

CE8   Please note that to avoid ambiguity, "/" was changed to "and", "or", "and/or", or "–". Please check the accuracy of these changes throughout.

CE9   Should this be a time? Or should it read "4–24 h"?

CE10   Please confirm.

CE11   Please note that language edits have been made to Figs. 2–5 and 9–10.

CE12   Please see above comment about italics in taxonomic names – can this be taken out of italics throughout (same with Hapyophyta)?

CE13   Are these titles of the pages? Or can the text and capitalization be adjusted?

CE14   Please confirm.

## Remarks from the typesetter

TS1   Please provide date of last access (dd/mm/yyyy).

TS2   Please provide date of last access (dd/mm/yyyy).

TS3   Please provide date of last access (dd/mm/yyyy).

TS4   Please note that units have been changed to exponential format throughout the text. Please check all instances.

TS5   Please provide date of last access (dd/mm/yyyy).

TS6   Please provide date of last access (dd/mm/yyyy).

TS7   Please confirm author name.

TS8   Is here something missing? Please check.

TS9   Please provide a reference list entry including creators, title, and date of last access.

TS10   Please provide a reference list entry including creators, title, and date of last access.

TS11   Please note that there is a discrepancy between funding information provided by you in the acknowledgements and the funding information you indicated during manuscript registration, which we used to create this section. Please double-check your acknowledgements to see whether repeated information can be removed from the acknowledgements or changed accordingly. If further funders should be added to this section, please provide the funder names and the grant numbers. Thanks.

TS12   Please confirm author name.

TS13   Please provide pages.

TS14   Please provide all authors.

TS15   Please provide pages.

TS16   Please provide date of last access (dd/mm/yyyy).

TS17   Please provide full journal name.

TS18   Please provide date of last access (dd/mm/yyyy).

TS19   Please provide pages.

TS20   Please provide pages.

TS21   Please provide journal name, volume and pages.

TS22   Please provide date of last access (dd/mm/yyyy).

TS23   Please provide pages.

TS24   Please provide pages.

TS25   Please provide date of last access (dd/mm/yyyy).

TS26   Please provide pages.

TS27    Please confirm.
TS28    Please provide pages.