# Peer review of "Waters and to Cumulative Chlorophyll-a in Anoxic Waters of"

_Biogeosciences, 2020_

## Referee Comment (RC1) · Anonymous Referee #1 · 20 Feb 2020

General comments

Several recent studies provided strong evidence of methane production in oxygenated freshwaters and seawater challenging the long-standing paradigm that microbial methane production occurs only under anoxic conditions and forces us to rethink the environmental dynamics of this greenhouse gas. Thus the manuscript by León-Palmero et al. certainly deals with one of the 'hot' scientific topics under current debate. The authors clearly show the occurrence of methane supersaturation in oxic surface waters (with seasonal dependence) of 12 reservoirs and discuss these results in the context of other environmental parameters such as abundance of photosynthetic

picoeukaryotes and cyanobacteria, and chlorophyll-a concentration. They found that dissolved methane was coupled to the abundance of photosynthetic picoeukaryotes during periods of summer stratification and the winter mixing, and to chlorophyll-a concentration and the abundance of cyanobacteria during the stratification period. Overall, this is an interesting and straightforward manuscript including novel results, but several issues need to be addressed before it is ready for publication.

Please note that I will not comment on the experimental set up of the DNA analysis and the respective results as this is not my particular field of expertise.

Specific comments

Throughout the whole manuscript including abstract: Please only show 3 significant digits for numbers presented throughout the manuscript. For example, presenting a number of "7082234" might be confusing and implies an analytical precision which is much better than you actually obtain. Furthermore, when large changes were observed you should better use "increased by orders of magnitude or by a factor of x to y".

Introduction: The literature review is fairly comprehensive. However, there are a few very recent studies (the authors might not be aware of) by Klintzsch et al. (2019) and Hartmann et al. (2020) dealing with formation of methane from picoeukaryotes and cyanobacteria in freshwater and seawater which the authors might include in the sections dealing with isotope studies and evidence for oxic methane production (this also applies to results and discussion section 3.2.2, "CH4-production coupled to photosynthetic organisms"). Furthermore, the study by Günthel et al. (2019) dealing with methane emissions with respect to lake sizes and the contribution of vertical and lateral methane transport should be mentioned and further discussed/included in section 3.2.2. "Vertical and lateral CH4-transport from anoxic environments".

page 3, line 65-66: Only the study by Damm et al. (2010) considered bacteria as a potential methane source. The other three studies investigated archaea as likely sources.

In some parts of the manuscript English writing could be improved (e.g. lines 26-31, 232-233, 277, 279-286, etc.). Please check carefully throughout the whole manuscript.

Material and Methods (2.1): I suggest showing a geographical map including all 12 reservoirs studied. This would help the reader to better envisage the geographical locations of all 12 reservoirs.

page 4, lines 140-143: "...NH4+ and NO2- concentrations..." . These data were neither shown nor discussed in the manuscript. Why?

page 6, section 2.5 DNA analysis: Please mention which samples were investigated for DNA analysis.

page 8, lines 220: Explain "V"!

page 9, section 3.2. and section 3.2.1.: Improve the flow between the two sections.

page 9, lines 269-272: Not very convincing argument. Please rewrite.

page 10 section 3. 2. 2. and subsection "Vertical and lateral CH4-transport from anoxic environments": Improve the flow between the two sections.

page 10, section 3. 2. 2.: For discussion please include and discuss recent results by Günthel et al. (2019).

page 11, line 300: "methylphosphonates (MPn)". If abbreviation was introduced before it is no more necessary to again use the full name and abbreviation. Please check throughout the whole manuscript for consistency.

Page 12, section "CH4-production coupled to photosynthetic organisms": As mentioned above there are a two very recent studies by Klintzsch et al. (2019) and Hartmann et al. (2020) which unambiguously show using stable isotope labeling approaches that both picoeukaryotes/phytoplankton such as Chrysochromulina sp., and cyanobacteria in freshwater and seawater produce methane per se. Please include these results in this section as they do fully support the results of the presented study.

Figures 1 to 7: Please provide more information about the statistical values of the parameters presented (e.g. error bars, SDs, uncertainty range, number of replicates, etc.). Furthermore, little information about the analytical uncertainties for the measurement systems is available in the method section. This needs to be improved in the revised manuscript.

Figures 1 to 3 and S1 to S9: Please add the date of sampling (field campaign) next to stratification period/mixing period.

Figure 2 legend, line 753: "The grey area represents the anoxic zone (DO < 7.5 $\mu$M)". There is no grey area highlighted in Figure 2.

Technical corrections

page 1, line 10, replace "CH4" by "methane"

page 1, line 28, include "more" after "much"

page 2, line 37: replace "called" by "described"

page 3, line 78: change to "...we considered the following CH4 sources:"

page 3, line 80: "León-Palmero et al. in review" has been nit listed in the reference section

page 4, line 103: spell out PAR (photo active radiation)

page 4, line 116: replace "concentration" by "mixing ratio"

page 9, line 268: delete "as free-living microorganisms"

Mentioned additional references:

Günthel, M., Donis, D., Kirillin, G., Ionescu, D., Bizic, M., McGinnis, D.F., Grossart, H.-P. and Tang, K.W. (2019) Contribution of oxic methane production to surface methane emission in lakes and its global importance. Nature Communications 10, 5497.

Hartmann, J.F., Günthel, M., Klintzsch, T., Kirillin, G., Grossart, H.-P., Keppler, F. and Isenbeck-Schröter, M. (2020) High Spatiotemporal Dynamics of Methane Production and Emission in Oxic Surface Water. Environ. Sci. Technol. 54, 1451-1463.

Klintzsch, T., Langer, G., Nehrke, G., Wieland, A., Lenhart, K. and Keppler, F. (2019) Methane production by three widespread marine phytoplankton species: release rates, precursor compounds, and potential relevance for the environment. Biogeosciences 16, 4129-4144.

---

## Referee Comment (RC2) · Anonymous Referee #2 · 26 Feb 2020

The manuscript "Dissolved CH4 coupled to Photosynthetic Picoeukaryotes in Oxic Waters and Cumulative Chlorophyll-a in Anoxia" by León-Palmero et al. presents CH4 measurements from the water column of 12 different reservoirs in southern Spain together with an assessment of different biological parameters, including the abundance of different plankton classes and specific functional genes that could indicate different CH4 production pathways. The sampling included seasonal measurements from the stratification period, when a pronounced thermal stratification separated the surface from the bottom waters, and the mixing period, when a uniform temperature profile throughout the water column was observed.

[Figure]

The authors clearly attributed the highest CH4 accumulation with highly oxygen-deficient waters, typically found in the bottom waters during the stratification period. They could not attribute in-situ production to gene abundances indicating archaeal methanogenenesis or methylphosphonate degradation, but they found a significant correlation in oxic waters between CH4 and a number of variables (the abundance of photosynthetic picoeukaryotes (PPEs), the mean depth of the reservoir, temperature and cyanobacteria abundance), across the different reservoirs.

The manuscript is generally well written and the results are consistent, but I am not fully convinced of the interpretation of the results. While I can agree with the authors' argumentation for the accumulation of CH4 in oxygen-deficient waters, I have some difficulties with the explanation for the CH4 distribution in the oxic waters where the authors attribute the CH4 distribution mainly to in-situ production by PPEs. Looking at the individual CH4 profiles, a correlation between PPEs and CH4 is not obvious, and I wonder if the overall significant correlation found by the authors rather reflects the variability between the reservoirs. Is there a significant correlation between PPEs and CH4 within the individual reservoirs?

I am furthermore not convinced that the vertical transport of CH4 plays a rather minor role for the CH4 distribution. The individual CH4 profiles seem to show the largest surface concentrations in reservoirs with a pronounced CH4 accumulation in the bottom waters (Type 1), which seems to indicate that vertical transport may indeed be an important source for CH4 in the surface.

The reservoirs' mean depth is a rather indirect proxy for vertical transport. Did the authors try to quantify the vertical flux based on the thermal stratification and the measured CH4 gradients? Are there other transport processes like ebullition or degassing that may introduce CH4 from the bottom waters to the surface? It would also be important to know how representative the sampling stations are for the entire reservoir. Can the authors give any information on the spatial variability within the reservoirs?

[Figure]

Specific comments:

Title: the title should state that the study is based on measurements from reservoirs.

Line 41: "CH4 inputs may become from..." replace "become" with "come"

Line 88: It would be good to have some additional information about the sampled reservoirs. A map showing the locations and shapes of the sampled reservoirs and the sampling location within the reservoirs would be very useful. What is the main purpose of the reservoirs? Are there human-induced parameters (e.g. periodic water discharge, nutrient input) that could impact the greenhouse gas budgets of the reservoirs? I think this information is necessary to understand the potential variablity across the reservoirs, particularly since the information given in the cited reference is in Spanish.

Line 89: replace "next" with "following"

Line 91: are reservoir volume and surface area constant variables? I can imagine that these numbers may show some variability.

Line 91: the description of mean depth calculation and equation (1) are somewhat redundant. I would either remove equation (1) or the description.

Line 105: Please give additional information about the water sampler. What is the volume and the closure mechanism of the sampler?

Line 111: What is the sampled volume for CH4 analysis and the relation between the sampled volume and the volume of the water sampler? Did the authors test for potential CH4 loss during the sampling procedure?

Line 280: replace "exportation" with "export"

Line 285: I would imagine that apart from their origin, the Chla content of the water column is more closely related to POM than DOM, so I am wondering if particulate organic matter (POM) would be more important for the CH4 production than DOM.

[Figure]

Figure 1-3 and S1-S9: while I found the general presentation of the individual reservoirs very useful, the partly logarithmic scale and the different scaling used for the individual profiles made the intercomparison of the data challenging. Maybe the authors could choose a uniform scaling for the profiles and use inserts to highlight the distribution where necessary.
* * *

---

## Referee Comment (RC3) · Anonymous Referee #3 · 27 Feb 2020

GENERAL COMMENTS:

Water column methane is commonly exclusively attributed to archaeal methanogenesis in anoxic sediments coupled with physical transport processes, especially in case of enclosed waterbodies. Throughout the last 2 decades evidence accumulated that methane can be also be produced under oxic conditions by archaeal and non-archaeal microbes. Some pathways have been identified, while others remain unknown. Withstanding, the relative contributions of the oxic and anoxic methane sources to whole-system budget are highly debated in the scientific community.

In their study, Leon-Palmero et al. investigate the origin of dissolved water column

methane in a series of reservoirs considering the commonly acknowledged sediment methanogenesis and the oxic methane source. The authors further resolve between different types of oxic methane sources: i) archaeal methanogenesis inside the water column, ii) microbial methylphosphonate degradation inside the water column, and iii) photosynthesis related methane production inside the water column. The authors apply a multi-method approach combining physicochemical analyses, gas analyses and molecular techniques to obtain a comprehensive dataset and proxy estimates for individual potential methane sources. The authors synthesize their results statistically to quantify the contribution of individual methane sources to dissolved water column methane. The following points summarize the major scientific advances of this study:

a) The oxic methane source(s) has not yet been investigated in reservoirs.

b) The study resolves the different methane sources throughout i) stratified, and ii) mixed period.

c) While the existences of the various methane sources have been reported in the literature, Leon-Palmero et al. present the first approach to resolve the contribution of anoxic sediment methanogenesis and various oxic sources simultaneously.

d) The general opinion in the scientific community is that either i) anoxic sediment methanogenesis, or ii) (oxic) methylphosphonate degradation are the dominant methane sources in whole basin budgets. In contrast, this study gives evidence that the photosynthetic methane source can be the dominant source in enclosed waterbodies what ties into the findings of several recent studies.

Accordingly, this study presents a series of new findings which is appropriate for Biogeosciences and will be interesting for a large readership. Applied methods are state of the art and clearly laid out, the authors are highly qualified. The authors present an appropriate introduction, however, the section about their research objectives falls a bit short (last introduction paragraph). As a result, the study appears less important to the reader than it actually is. I suggest, better connecting the introduction with the

research objectives (e.g. recalling the scientific dispute of oxic versus anoxic methane sources and highlighting the major advances of the study – general advances will fit this paragraph whereas details may be stated in the conclusion section).

The authors should, further, include an overview of reservoir characteristics (carbon/phosphorus/nitrogen, trophic state, surface area, shoreline, mean depth, depth of the mixed layer). Also sampling locations should be characterized in a more detailed manner (location within the reservoir, shore distance, depth), for example, in tabular form. Towards the end of the Result&Discussion section I recommend recalling the general dispute between oxic and anoxic contribution and implementing the statistical results. Yet, there are not many literature information available, but please, place your overall finding into the context of DelSontro et al. 2018 and Günthel et al. 2020 (only studies so far presenting/summarizing contribution patterns for a series of lakes) who presented evidence for the importance of the oxic source increasing with basin size.

Does the dataset indicate reservoir conditions favoring individual oxic methane sources?

On some occasions the English should be improved throughout which will lead to a better understanding by the readership. Throughout the specific comments I acknowledged some of these occasions with rephrasing statements.

Statistical correlations make up a huge part of the study. I would find it practical to include a table summarizing the outcome of all significant and non-significant correlations (stating type of test, predictor variables and effect, p-value; covering not only the GAM model results). It will be helpful, to reference display items like this table in the method section (in this regard please check all the Supplementary Materials). Also, please include the correct units when statistical equations are stated throughout. Further, please check the wording when describing statistical results – on several occasions the statistical results are described with phrases like 'explains', 'determines' etc. Please use wording like 'points to explain', 'may determine', 'appears to be a main

driver' etc. instead. Although the correlations appear to partially explain a lot of variance, no actual (production) rates have been measured, and this should be reflected by the language.

The authors present a comprehensive dataset including many parameters. Please describe the obtained data in more detail throughout the method section or when recalled by display items. While many of the conditions can be read out of other display items, it complicates the reading flow. For example, in Fig. 7, data points are grouped into stratified and mixed data. The reader has no information about corresponding depths. But different conditions in epilimnic versus hypolimnic waters may lead to different contribution patterns of individual methane sources. Did the authors check if splitting the dataset into epilimnic/hypolimnic data improves the correlations? Another example are the PCR results presented in Figs. S11 and S12. What reservoirs (depths, time points, replicates?) are resembled by samples 1-12? Please clarify these details.

In summary, the authors present a valuable study for the scientific community. I recommend publication after addressing the general and specific comments.

SPECIFIC COMMENTS:

Abstract:

L 10-11. Please define CH4.

L 15-16. Methane supersaturation is a common observation in aquatic systems, including oxic waters (e.g. Tang et al. 2016 and references herein). Without differentiating between oxic and anoxic methane sources this sentence adds little content (e.g. close to sediments/in anoxia methane supersaturation is generally expected). Rephrase or combine with the following sentence. I recommend avoiding the percentage numbers, especially if they do not refer to significant correlations with picoeukaryotes or cumulative chlorophyll-a.

L 16-17. Here, it is important to state the size of the study reservoirs.

L 19. Replace 'determined' with a more appropriate wording, e.g. 'correlated with' etc.

L 20-21. Please rephrase this sentence.

Introduction:

L 77-83. Explain your research questions in more detail and state why they are of interest (link them better to the previous introductory part). - Picoeukaryotes are first mentioned in the (too short) section describing research questions, not all readers are familiar with this terminology. Accordingly, it should be defined. What organism does it include? Why are they of interest within your research agenda - Actual methane production rates have not been measured but were deduced from proxies!

L 28. Change 'much' to 'more'

L 29. Change 'attributed to' to 'determined by'

L 36-37. Add references, e.g. Tang et al. (2016) + references herein

L 41. Rephrase 'CH4 inputs may become from', e.g. 'CH4 may originate from'

L 44-47. Please incorporate the findings by Thalasso et al. (2020)

L 47-48. Please reference the findings by DelSontro et al. (2018)

L 52-53. Reference Hartmann et al. (2020), Bizic et al. (2020a) (isotopes); Khatun et al. (2020), Yao et al. (2016) (molecular approaches)

L 62. Change 'contrary', e.g. to 'in contrast'

L 78. Diverse in what sense? Please clarify.

Methods:

L 85-91. Please give an overview on the general reservoir characteristics (trophic state, size, location, temperature and oxygen conditions, epilimnion/hypolimnion depths etc). The reader requires this information to better place presented results into the study

context. Searching for all parameters in various figures hampers a direct comparison. In the following a series of different measurements are described. However, it is unclear if all these measurements were done simultaneously (at the same day or week) and at the same sites. Please clarify (maybe as a Supplementary Table). When measurements were done during mixed and stratified periods, have the exact same sites been re-sampled (location, depth, shore distance)?

L 88-90. Please rephrase this sentence.

L 91-93. Decide for either equation 1 or a definition in the running text to reduce redundancy.

L 102-104. Please state the units of each parameter as used throughout, and clarify what fluorescence relates to.

L 104-105. Do you mean measurement intervals in 6 or 9 m steps? Please clarify.

L 148-152. Please clarify the difference between integrated mean (integrated over depth? Consider defining it as total amount/content.), and cumulative chlorophyll a concentration. Consider labeling them with different Symbols.

L 168. Please state the equations used to compute GPP, NEP and R.

L. 180-181. Please clarify what you mean by 'specific primers from similar studies'. What pure cultures did you use (type, culture conditions, origin)?

L 170-196. Please lay out when (in the context of other parameters) and where (location in the reservoir and depth) you sampled. Also, specify which reservoirs have been sampled (you state 12 reservoirs throughout section 2.1, but Fig. S11 only shows 10 samples). Please clarify, what samples have been measured (depth, mixed/stratified period). Also, please add this information to Fig. S11 and S12 (define 'samples 1-12').

Sample 6 in Fig. S12 appears to have a very weak signal close to the phnJ bar. Did you verify that this is no positive signal (e.g. edit light/contrast properties to better resolve

this area)? What was used as negative controls (in both assays: mcrA, phnJ)?

L 194-196. What was the DNA range you investigated?

Results&DISCUSSION:

L 218-219. Please avoid the large and unpractical percentage numbers (hard to read). Given, that the authors state the dissolved concentration, no content is lost when removing these numbers. For example, the authors could incorporate the average saturation concentrations in the following sentence.

L 223-225. The references are mixed up. E.g., Donis et al., Grossart et al., Tang et al. investigated temperate lakes, but Murase et al. researched a tropical lake etc. Please rephrase this sentence.

L 225. Please define the depth of 'surface waters'. Why is emphasize placed on Lake Kivu (not listed throughout the previous sentence)?

L 248. Please mention that the literature refers to studies in lakes. Note, a metalimnic methane maximum can be controlled by physical (e.g. differential gas solubility due to temperature change, emission) and biological factors (e.g. light inhibition of methane oxidation, variable phytoplankton methane production due to availability of nutrients/light/precursors). Also note, Kathun et al. (2019) presented a series of lakes with and without metalimnic methane maxima.

L 254. Referenced literature is not about boreal lakes.

L 256-263. Reference Tang et al. (2014) who also presented a distinction between oxic and anoxic methane concentrations in oxic and anoxic lake waters.

L 264-269. Please indicate what depths you analyzed.

L 272-273. Please rephrase the sentence. Also, please emphasize archaeal methane production is absent in the anoxic water column.

L 282. Please reword 'depend'. E.g. correlate with.

L 289-291. Please reference these considerations/results here.

L 296. Please define 'extreme' P limitation. Different Organisms require different amount of P. Accordingly, this is a relative terminology. Does it relate to the N:P ratio? Please clarify.

L 299-304. I appreciate the authors approach to evaluate the contribution of anoxic methane based on morphometrical parameters. However, lateral transport which is seen as major source of epilimnic methane is modulated by the shore-mid distance. Accordingly, when correlations are done this should be accounted for. Please clarify shore-mid distances in your dataset. I think, Hofmann et al. 2010 and DelSontro et al. 2018 should be referenced as these studies show major contributions by the littoral (especially in smaller lakes).

L 304-305. Please reference the lateral transport model by DelSontro et al. (2018) which agrees with the observation of exponential decay functions. A later discussion on this decay function should be warrant. E.g., close to the shore there should be a high content of dissolved methane that originated from the sediments; with increasing distance from the shore the relative contribution of the oxic methane source should increase.

L 306-307. Concentrations increased exponentially versus what parameter?

L 313-314. Please discuss why there was no significant correlation between dissolved methane and shallowness index. Potential reasons might be: variable sediment methane production rates among reservoirs (variable among temperature, trophic state, sediment porosity, soil type and community etc.) the ratio between oxic and anoxic methane production may lead to distortion. The authors may have other points. Did you try the ratio of 'mean depth : depth of surface mixed layer' as a proxy for lateral input (depth of the mixed layer relates to the amount of temperate sediments)?

L 315-317. I think the authors should remove the wind speed correlation throughout. Wind-forcing, or more generally, turbulence which drives surface emission and substantially affects surface water methane concentrations, is modulated by many environmental factors (basin geomorphometry, temperature, rain etc.). Also, internal production rates, lateral methane input and methane oxidation modulate surface concentrations beside emission. Accordingly, correlating wind speed with surface concentrations is an over-simplification and might be misleading.

L 317-318. Please rephrase.

L 318-319. Please define what you mean by 'extreme supersaturation' and where these concentrations were found.

L 323-327. This is an interesting observation. Consider highlighting this observation more. Is it possible that you filtered off the attachment partners of methanogenic Archaea (you filtered water samples before molecular analyses)?

L 341-342. Define 'extreme' limitation. According to my knowledge, it is unknown what are the P levels (N:P ratio) triggering MPN degradation and corresponding methane production in the field.

L 342-343. Are these correlations simple x-y regressions or do they include for multiple predictor variables?

L 345-350. Fig S12. Can you change the picture properties (light, contrast); it seems, there is a faint bluer in sample 6. Cyanobacteria have been detected (e.g. Fig. S13). What type of cyanobacteria were present? Do these result agree with findings by Yao et al. 2016 (some types possess the methane generating enzyme machinery)?

L 352-364. Many information listed belong to the result/method section. Please remove redundancy.

L 365-366. Given that many factors may affect the dissolved methane concentration (as discussed throughout) a p-value of 0.077 might still point to a connection. It has

been stated n=12; does that mean 1 value per reservoir during the stratified period? Same questions on other occasions. Please clarify.

383-384. Please add the reference Hartmann et al. 2020 who presented methane production by green algae, cyanobacteria, cryptophytes and diatoms.

L 396-405. Please mention methylated amines which can also serve as methane precursors (e.g. Bizic et al. 2018, Bizic et al. 2020b).

L 412-414. Please emphasize that this cyanobacterial methane 'production' does here not relate to MPN degradation (following the absence of phnJ).

L 435-436. Please rephrase and avoid the percentage number. Results by Bogard et al. (2014), Donis et al (2017), DelSontro et al. (2018) and Günthel et al. (2019) suggest that in larger waterbodies the majority of surface mid-water methane (/emission) might be explained by the oxic source.

L 437-438. Please rephrase.

L 458-460. I suggest moving the PPEs description to the result section or even earlier, when the terminology PPE is defined.

L 461-465. Please indicate to what extent the different methane sources might explain the dissolved methane concentrations (e.g. in percentage – relates to Fig. 8a).

L 467-468. Given other work on relationships between methane and chlorophyll/non-cyanobacteria phytoplankton (Tang et al. 2016, Hartmann et al. 2020), I think novel is not the right terminology here.

L 475-476. Please rephrase.

References:

Please check abbreviation punctuation (sometimes with and sometimes without dot).

L 582. Typo.

L 627. Capital letters

DISPLAY ITEMS:

L 760-761. Please state the regression statistics in the figure legend.

L 765-767. Please state the regression statistics in the figure legend.

L 769-771. Did you have data about reactive phosphorus? Using biologically accessible phosphorus instead of total phosphorus might improve the correlation statistics.

L 775-780. In case R2<1.0, the functions do not entirely explain the dissolved methane concentration data but only a certain fraction of data variance (e.g. 40% at R2=0.40 in a simple regression). Please rephrase accordingly.

What water depths do these readings (n=78 or 82) correspond to? In case some belong to the epilimnion and some to the hypolimnion, different nutrient availabilities may affect the correlation statistics and mask potential relationships.

L 784. (corresponding to Fig. 8a). I am not sure, 'significance' is the best terminology at this point. Consider rephrasing y-axes to explanatory power.

Could the author please comment on the partial effect of 'sediment methane production' (Fig. 8b)? Why is the trendline reversing the slope after ca. 27.5m mean depth? Does this parameter indicate bigger (deeper) reservoirs than 27.5 m mean depth have a higher sediment contribution to mid-water methane?

Supplementary Materials:

Table S1. Please discuss why the statistical correlation leads to substantial differences when applied to the combined data set of stratified and mixed period (e.g. no significant correlation listed with mean depth what is here used as a proxy for conventional sediment methanogenesis). Please discuss these details in the main document.  
REFERENCES:

Bižić-Ionescu M., Ionescu D., Günthel M., Tang K.W., Grossart HP. (2018) Oxic Methane Cycling: New Evidence for Methane Formation in Oxic Lake Water. In: Stams A., Sousa D. (eds) Biogenesis of Hydrocarbons. Handbook of Hydrocarbon and Lipid Microbiology. Springer, Cham

Bižić, M., Klintsch, T., Ionescu, D., Hindiyeh, M. Y., Günthel, M. et al. Aquatic and terrestrial cyanobacteria produce methane. Sci. Adv. 6, eaax5343 (2020a). doi.10.1126/sciadv.aax5343

Bižić, M., Grossart, H.‐P. and Ionescu, D. (2020b). Methane Paradox. In eLS, John Wiley & Sons, Ltd (Ed.). doi:10.1002/9780470015902.a0028892

Bogard, M., del Giorgio, P., Boutet, L. et al. Oxic water column methanogenesis as a major component of aquatic CH4 fluxes. Nat Commun 5, 5350 (2014). https://doi.org/10.1038/ncomms6350

DelSontro, T., del Giorgio, P.A. & Prairie, Y.T. No Longer a Paradox: The Interaction Between Physical Transport and Biological Processes Explains the Spatial Distribution of Surface Water Methane Within and Across Lakes. Ecosystems 21, 1073–1087 (2018). https://doi.org/10.1007/s10021-017-0205-1Hartmann et al. 2020

Günthel, M., Donis, D., Kirillin, G. et al. Contribution of oxic methane production to surface methane emission in lakes and its global importance. Nat Commun 10, 5497 (2019). https://doi.org/10.1038/s41467-019-13320-0

Khatun, S.; Iwata, T.; Kojima, H.; Ikarashi, Y.; Yamanami, K.; Imazawa, D.; Kenta, T.; Shinohara, R.; Saito, H. Linking Stoichiometric Organic Carbon–Nitrogen Relationships to planktonic Cyanobacteria and Subsurface Methane Maximum in Deep Freshwater Lakes. Water 2020, 12, 402.

Khatun, S., Iwata, T., Kojima, H., Fukui, M., Aoki, T. et al. Aerobic methane production by planktonic microbes in lakes. Sci. Tot. Environ. 696, 133916 (2019). doi.org/10.1016/j.scitotenv.2019.133916

Tang, Kam W., McGinnis, Daniel F., Frindte, Katharina, Brüchert, Volker, Grossart, Hans-Peter. Paradox reconsidered: Methane oversaturation in well‐oxygenated lake waters, Limnology and Oceanography, 59 (2014). doi: 10.4319/lo.2014.59.1.0275.

Tang, K. W., McGinnis, D. F., Ionescu, D. and Grossart, H. P. Environmental Science & Technology Letters 2016 3 (6), 227-233 DOI: 10.1021/acs.estlett.6b00150

Thalasso, F., Sepulveda-Jauregui, A., Gandois, L. et al. Sub-oxycline methane oxidation can fully uptake CH4 produced in sediments: case study of a lake in Siberia. Sci Rep 10, 3423 (2020). https://doi.org/10.1038/s41598-020-60394-8

Yao, M., Henny, C. and Maresca, J. A. Freshwater Bacteria Release Methane as a By-Product of Phosphorus Acquisition. Appl. Environ. Microbiol. 82(23): 6994-7003 (2016).
* * *

---

## Author Comment (AC1) · 8 Apr 2020

Please see below the point-by-point responses to the reviewers and the actions taken regarding their concerns. In the text below, the suggestions and comments of the reviewers are in black and plain font, and our responses are in italics and blue font.

Anonymous Referee #2

**COMMENTS TO THE AUTHOR (S)**
The manuscript is generally well written and the results are consistent, but I am not fully convinced of the interpretation of the results. While I can agree with the authors' argumentation for the accumulation of CH4 in oxygen-deficient waters, I have some difficulties with the explanation for the CH4 distribution in the oxic waters where the authors attribute the CH4 distribution mainly to in-situ production by PPEs. Looking at the individual CH4 profiles, a correlation between PPEs and CH4 is not obvious, and I wonder if the overall significant correlation found by the authors rather reflects the variability between the reservoirs. Is there a significant correlation between PPEs and CH4 within the individual reservoirs?

*In this manuscript, we concluded that the dissolved $CH_4$ concentration in oxic waters results from several non-exclusive sources as the vertical transport in shallow reservoirs, temperature, and in situ production by photosynthetic picoeukaryotes (PPEs) and cyanobacteria. The abundance of PPEs explained the largest part of the variance in the dissolved $CH_4$ in the GAM model for the twelve reservoirs during the stratification period (Figure 9), and was significant, along with mean depth, during mixing (Figure 10). The variance explained by each driver may vary among reservoirs when they are analyzed individually, because of intrinsic reservoir properties and a reduction in the range of variability. Therefore, the correlation between PPEs and $CH_4$ may not appear obvious in all the profiles. The scale variability of this work was across reservoir typologies, not within reservoirs. However, despite that within reservoirs was not the scale of this study, we found a significant and direct correlation between PPEs and the dissolved $CH_4$ in the next reservoirs: Jándula (p-value < 0.01), Los Bermejales (p-value < 0.05), Francisco Abellán (p-value < 0.01), El Portillo (p-value < 0.01), and Colomera (p-value < 0.001).*

I am furthermore not convinced that the vertical transport of CH4 plays a rather minor role for the CH4 distribution. The individual CH4 profiles seem to show the largest surface concentrations in reservoirs with a pronounced CH4 accumulation in the bottom waters (Type 1), which seems to indicate that vertical transport may indeed be an important source for CH4 in the surface. The reservoirs' mean depth is a rather indirect proxy for vertical transport. Did the authors try to quantify the vertical flux based on the thermal stratification and the measured CH4 gradients? Are there other transport processes like ebullition or degassing that may introduce CH4 from the bottom waters to the surface? It would also be important to know how representative the sampling stations are for the entire reservoir. Can the authors give any information on the spatial variability within the reservoirs?

*The vertical transport did not play a minor role in the dissolved $CH_4$ distribution. In the manuscript, we have explicitly shown that the mean depth (i.e., surrogate of vertical*

*transport) was significantly related to the dissolved $CH_4$ concentration in oxic waters both during the stratification (Figure 9) and during the mixing (Figure 10). According to the results of the Generalized additive models (GAMs), vertical transport was the second driver in importance explaining the dissolved $CH_4$ in the oxic waters during the stratification period and the first one during the mixing period.*

*We think that other processes as ebullition or degassing that might introduce $CH_4$ from the bottom waters to the surface, but we did not measure the vertical flux based on the thermal stratification and $CH_4$ gradients. We considered that the mean depth is a worthy, easy to obtain proxy for the vertical transport of $CH_4$. Unfortunately, we did not study the spatial variability within the reservoirs. The target scale of this work was across-reservoir variability during the stratification and the mixing period. Within-reservoir spatial variability would require a more detailed study maybe just in one or two reservoirs, but hardly feasible in 12 reservoirs.*

**Specific comments:**
Title: the title should state that the study is based on measurements from reservoirs.

*We have included the word "reservoirs" in the title.*

Line 41: "CH4 inputs may become from..." replace "become" with "come"
Line 89: replace "next" with "following"

*We replaced these words in the manuscript (Lines 42 and 103).*

Line 88: It would be good to have some additional information about the sampled reservoirs. A map showing the locations and shapes of the sampled reservoirs and the sampling location within the reservoirs would be very useful. What is the main purpose of the reservoirs? Are there human-induced parameters (e.g. periodic water discharge, nutrient input) that could impact the greenhouse gas budgets of the reservoirs? I think this information is necessary to understand the potential variability across the reservoirs, particularly since the information given in the cited reference is in Spanish.

*According to the reviewer´s suggestions, we have included a new figure (Figure 1) and two tables in the revised version of the manuscript to describe the study reservoirs. Figure 1 shows the geographical location of the reservoirs, and in Table 1 we have included the geographical coordinates, the year of construction, and the morphometric parameters of the reservoirs. In Table 2 we have included basic reservoir characteristics: carbon/phosphorus/nitrogen, and chlorophyll-a concentrations*
*The main purpose that led to the construction of these reservoirs was the water supply and agriculture irrigation (Lines 97-98). We have also included a recently published reference of a study performed in these twelve reservoirs (León-Palmero et al. 2020). In this publication, we studied the importance of the watershed on the emissions of greenhouse gases and the information for each reservoir is very detailed there.*

Line 91: are reservoir volume and surface area constant variables? I can imagine that

these numbers may show some variability.

*Reservoir volume and surface area are not constant variables. However, in this work, we studied a very heterogeneous group of reservoirs, and we assumed that within-reservoir variability was less critical than across-reservoir variability. Therefore, we assumed that the values obtained from the general dimensions were representative of such heterogeneity.*

Line 91: the description of mean depth calculation and equation (1) are somewhat redundant. I would either remove equation (1) or the description.

*We removed the description and kept the equation (1) (Line 107).*

Line 105: Please give additional information about the water sampler. What is the volume and the closure mechanism of the sampler?

*We took the water samples using a UWITEC sampling bottle of 5 liters of capacity. The water sampler was self-closing (Lines 121-122).*

Line 111: What is the sampled volume for CH4 analysis and the relation between the sampled volume and the volume of the water sampler? Did the authors test for potential CH4 loss during the sampling procedure?

*The water sampler has 5 liters of capacity, and we filled three 125 mL Winkler bottles or two 250 mL Winkler bottles. We measured dissolved $CH_4$ using headspace equilibration in a 50 ml air-tight glass syringe by duplicate (in 250 mL bottles) or triplicate (in 125 mL bottles) from each Winkler bottle. We took a quantity of 25 g of water (± 0.01 g) using the air-tight syringe. Therefore, the sampled volume for $CH_4$ analysis in the Winkler bottles is 7.5 % - 10 % of the total bottle volume.*
*We tried to minimize the $CH_4$ loss by filling up the Winkler bottles very carefully from the bottom to avoid the formation of bubbles and sealing the Winkler bottles with Apiezon® grease to prevent gas exchange.*

Line 280: replace "exportation" with "export"

*We did the correction suggested by the reviewer (Line 359).*

Line 285: I would imagine that apart from their origin, the Chla content of the water column is more closely related to POM than DOM, so I am wondering if particulate organic matter (POM) would be more important for the $CH_4$ production than DOM.

*Usually, the POM pool more dynamics than the DOM pool and is well correlated to chlorophyll-a, but also exopolymers released by phytoplankton and bacteria contribute significantly to POM pool and export. In the study reservoirs, the dissolved organic carbon concentration was significantly related to the age of the reservoirs and the forestry area in their watersheds (León-Palmero et al., 2019). Therefore, in terms of quality, the total DOM may represent the allochthonous, aged and more resistant to*

*microbial degradation of the carbon pool. In contrast, the autochthonous organic matter derived from phytoplankton may represent a labile and biodegradable fraction (Lines 363-367).*

Figure 1-3 and S1-S9: while I found the general presentation of the individual reservoirs very useful, the partly logarithmic scale and the different scaling used for the individual profiles made the intercomparison of the data challenging. Maybe the authors could choose a uniform scaling for the profiles and use inserts to highlight the distribution where necessary.

*We agreed with the reviewer that the inter-comparison of the data might be challenging because of the different scaling. We used the same scale for water temperature and dissolved oxygen concentration. However, the dissolved $CH_4$ concentration among reservoirs ranged up to four orders of magnitude (0.02-213.64 $\mu M$), and to be able to see differences among reservoirs was impossible to use, unfortunately, the same scale for dissolved $CH_4$.*

**References**

León-Palmero, E., Reche, I. and Morales-Baquero, R.: Atenuación de luz en embalses del sur-este de la Península Ibérica, Ingeniería del agua, 23(1), 65–75, doi:10.4995/ia.2019.10655, 2019.

León-Palmero, E., Morales-Baquero, R. and Reche, I.: Greenhouse gas fluxes from reservoirs determined by watershed lithology, morphometry, and anthropogenic pressure, Environ. Res. Lett., 15(4), 044012, doi:10.1088/1748-9326/ab7467, 2020.

---

## Author Comment (AC2) · 8 Apr 2020

Please see below the point-by-point responses to the reviewers and the actions taken regarding their concerns. In the text below, the suggestions and comments of the reviewers are in black and plain font, and our responses are in italics and blue font.

Anonymous Referee #1

COMMENTS TO THE AUTHOR (S)
Throughout the whole manuscript including abstract: Please only show 3 significant digits for numbers presented throughout the manuscript. For example, presenting a number of "7082234" might be confusing and implies an analytical precision which is much better than you actually obtain. Furthermore, when large changes were observed you should better use "increased by orders of magnitude or by a factor of x to y".

*We have checked the whole manuscript and followed the reviewer suggestion.*

Specific comments

Introduction.
The literature review is fairly comprehensive. However, there are a few very recent studies (the authors might not be aware of) by Klintzsch et al. (2019) and Hartmann et al. (2020) dealing with formation of methane from picoeukaryotes and cyanobacteria in freshwater and seawater which the authors might include in the sections dealing with isotope studies and evidence for oxic methane production (this also applies to results and discussion section 3.2.2, "CH4-production coupled to photosynthetic organisms"). Furthermore, the study by Günthel et al. (2019) dealing with methane emissions with respect to lake sizes and the contribution of vertical and lateral methane transport should be mentioned and further discussed/included in section 3.2.2. "Vertical and lateral CH4-transport from anoxic environments".

*We agree with the reviewer that these articles are relevant for this manuscript, but we were not aware of the existence of these three recent studies at the moment of sending the manuscript. We thank reviewer advice and have included these references in the revised manuscript in the Introduction (lines 58, and 81) and Results and Discussion sections.*

page 3, line 65-66: Only the study by Damm et al. (2010) considered bacteria as a potential methane source. The other three studies investigated archaea as likely sources.

*We rephrased the sentence in the revised manuscript as follows:*
*"In the open ocean, archaea and bacteria appear to metabolize the algal osmolyte dimethylsulfoniopropionate producing methane as a byproduct (Damm et al., 2008, 2010, 2015; Zindler et al., 2013)" (Lines 72-73).*

In some parts of the manuscript English writing could be improved (e.g. lines 26-31, 232-233, 277, 279-286, etc.). Please check carefully throughout the whole manuscript.

*We have checked the English writing throughout the whole manuscript and correctly rewrote the parts suggested by the reviewer.*

Material and Methods
(2.1): I suggest showing a geographical map including all 12 reservoirs studied. This would help the reader to better envisage the geographical locations of all 12 reservoirs.

*We have included a new figure that shows the location of the study reservoirs (new Figure 1). We have also included geographical coordinates and ancillary information in the new Table 1 and Table 2.*

page 4, lines 140-143: "NH4+ and NO2- concentrations: : :" . These data were neither shown nor discussed in the manuscript. Why?

*In the first version of this manuscript, we used the nitrate, nitrite and ammonia concentration to calculate the concentration of dissolved inorganic nitrogen (DIN). We used the DIN: TP ratio as an index for the phosphorus limitation, but the information was not explicitly explained. Now, in the revised manuscript, we have included the information how we obtain DIN in the method section (lines 170-171) and a table with basic information about the concentrations of carbon, nitrogen, phosphorus and the ratio DIN: TP in the water column of the study reservoirs (Table 2).*

page 6, section 2.5 DNA analysis: Please mention which samples were investigated for DNA analysis.

*We have clarified the details of the DNA analysis in the Method section "2.5 DNA analysis". From the sampling of the water column, we selected 3 or 4 representative depths for the epilimnion, metalimnion (oxycline), and hypolimnion/bottom layers to determine the abundance of the functional genes during the stratification period. We also selected 3 or 4 equivalent depths during the mixing period. In total, we analyzed 77 samples: 41 samples from the stratification period, and 36 samples from the mixing period. We have included this information in the revised manuscript (Lines 234 – 237).*

page 8, lines 220: Explain "V"!

*We performed the Wilcoxon signed rank-sum test to compare the concentration and the saturation of $CH_4$ between the stratification and the mixing period. V is the statistical parameter obtained from this test and we have included this information in the method section (Line 272). We used the Wilcoxon test because our data were not normally distributed.*

page 9, section 3.2. and section 3.2.1.: Improve the flow between the two sections.

*We have rewritten the connection between the two sections (Line 339).*

page 9, lines 269-272: Not very convincing argument. Please rewrite.

*We have rewritten the argument in the revised manuscript (Lines 344-353).*

page 10 section 3. 2. 2. and subsection "Vertical and lateral CH4-transport from anoxic environments": Improve the flow between the two sections.

*We have slightly rewritten the connection between these two sections (lines 377-382).*

page 10, section 3. 2. 2.: For discussion please include and discuss recent results by Günthel et al. (2019).

*We have included the results of Günthel et al. (2019) in the discussion of the revised manuscript (Lines 540-542).*

page 11, line 300: "methylphosphonates (MPn)". If abbreviation was introduced before it is no more necessary to again use the full name and abbreviation. Please check throughout the whole manuscript for consistency.

*We have checked and corrected it throughout the whole manuscript.*

Page 12, section "CH4-production coupled to photosynthetic organisms": As mentioned above there are a two very recent studies by Klintzsch et al. (2019) and Hartmann et al. (2020) which unambiguously show using stable isotope labeling approaches that both picoeukaryotes/phytoplankton such as Chrysochromulina sp., and cyanobacteria in freshwater and seawater produce methane per se. Please include these results in this section as they do fully support the results of the presented study.

*We agree with the reviewer, these two studies fully support our results. Therefore, we thank reviewer´s advice and we have included them in the revised manuscript (Lines 476- 479 and 486).*

Figures 1 to 7: Please provide more information about the statistical values of the parameters presented (e.g. error bars, SDs, uncertainty range, number of replicates, etc.). Furthermore, little information about the analytical uncertainties for the measurement systems is available in the method section. This needs to be improved in the revised manuscript.

*Throughout the methods section, we have included more information about the replicates, statistical values of the parameters presented and analytical uncertainties. Please, see the method section (lines 141, 143, 165, 166, 169, 171, 174, 181).*

Figures 1 to 3 and S1 to S9: Please add the date of sampling (field campaign) next to stratification period/mixing period.

*We have included the dates of sampling in the figure captions, and in the new Table 2.*

Figure 2 legend, line 753: "The grey area represents the anoxic zone (DO < 7.5 _M)". There is no grey area highlighted in Figure 2.

*That sentence was included by mistake; therefore in the revised manuscript we have removed it.*

Technical corrections
page 1, line 10, replace "CH4" by "methane"
page 1, line 28, include "more" after "much"
page 2, line 37: replace "called" by "described"
page 3, line 78: change to ": we considered the following CH4 sources:"
page 3, line 80: "León-Palmero et al. in review" has been nit listed in the reference section
page 4, line 103: spell out PAR (photo active radiation)
page 4, line 116: replace "concentration" by "mixing ratio"
page 9, line 268: delete "as free-living microorganisms"

*We did most of the technical corrections suggested by the reviewer. We also replaced the reference "León-Palmero et al. in review" by "León-Palmero et al. (2020)" throughout the manuscript, and now this reference is listed in the reference section.*

*Concerning the last point, we have decided to keep "as free-living microorganism". We consider that it is necessary to clarify that the negative results in the analysis of the mcrA gene in the water column of the reservoirs mean that we did not find methanogenic Archaea in the water column as "as free-living microorganisms". They could have been present in anoxic micro-niches (i.e., in the guts of zooplankton or within sinking particles) but these other potential sources were not analyzed in this study. Previous works have demonstrated the occurrence of methanogenesis in micro-anoxic zones of zooplankton guts and within sinking particles (Angelis and Lee, 1994; Karl and Tilbrook, 1994). We have included this explanation in the main text (Lines 418-420).*

**References:**

Angelis, M. A. de and Lee, C.: Methane production during zooplankton grazing on marine phytoplankton, Limnol. Oceanogr., 39(6), 1298–1308, doi:10.4319/lo.1994.39.6.1298, 1994.

Damm, E., Kiene, R. P., Schwarz, J., Falck, E. and Dieckmann, G.: Methane cycling in Arctic shelf water and its relationship with phytoplankton biomass and DMSP, Mar. Chem., 109(1–2), 45–59, doi:10.1016/j.marchem.2007.12.003, 2008.

Damm, E., Helmke, E., Thoms, S., Schauer, U., Nöthig, E., Bakker, K. and Kiene, R. P.: Methane production in aerobic oligotrophic surface water in the central Arctic Ocean, Biogeosciences, 7(3), 1099–1108, doi:10.5194/bg-7-1099-2010, 2010.

Damm, E., Thoms, S., Beszczynska-Möller, A., Nöthig, E. M. and Kattner, G.: Methane excess production in oxygen-rich polar water and a model of cellular conditions for this paradox, Polar Sci., 9(3), 327–334, doi:10.1016/j.polar.2015.05.001, 2015.

Karl, D. M. and Tilbrook, B. D.: Production and transport of methane in oceanic particulate organic matter, Nature, 368(6473), 732–734, doi:10.1038/368732a0, 1994.

León-Palmero, E., Morales-Baquero, R. and Reche, I.: Greenhouse gas fluxes from reservoirs determined by watershed lithology, morphometry, and anthropogenic pressure, Environ. Res. Lett., 15(4), 044012, doi:10.1088/1748-9326/ab7467, 2020.

Marotta, H., Pinho, L., Gudasz, C., Bastviken, D., Tranvik, L. J. and Enrich-Prast, A.: Greenhouse gas production in low-latitude lake sediments responds strongly to warming, Nature Clim. Change, 4(6), 467–470, doi:10.1038/nclimate2222, 2014.

Sepulveda-Jauregui, A., Hoyos-Santillan, J., Martinez-Cruz, K., Walter Anthony, K. M., Casper, P., Belmonte-Izquierdo, Y. and Thalasso, F.: Eutrophication exacerbates the impact of climate warming on lake methane emission, Sci. Total Environ., 636, 411–419, doi:10.1016/j.scitotenv.2018.04.283, 2018.

Yvon-Durocher, G., Allen, A. P., Bastviken, D., Conrad, R., Gudasz, C., St-Pierre, A., Thanh-Duc, N. and del Giorgio, P. A.: Methane fluxes show consistent temperature dependence across microbial to ecosystem scales, Nature, 507(7493), 488–491, doi:10.1038/nature13164, 2014.

Zindler, C., Bracher, A., Marandino, C. A., Taylor, B., Torrecilla, E., Kock, A. and Bange, H. W.: Sulphur compounds, methane, and phytoplankton: interactions along a north–south transit in the western Pacific Ocean, Biogeosciences, 10(5), 3297–3311, doi:10.5194/bg-10-3297-2013, 2013.

---

## Author Comment (AC3) · 8 Apr 2020

Please see below the point-by-point responses to the reviewers and the actions taken regarding their concerns. In the text below, the suggestions and comments of the reviewers are in black and plain font, and our responses are in italics and blue font.

Anonymous Referee #3

COMMENTS TO THE AUTHOR (S)
Applied methods are state of the art and clearly laid out, the authors are highly qualified. The authors present an appropriate introduction, however, the section about their research objectives falls a bit short (last introduction paragraph). As a result, the study appears less important to the reader than it actually is. I suggest, better connecting the introduction with the research objectives (e.g. recalling the scientific dispute of oxic versus anoxic methane sources and highlighting the major advances of the study – general advances will fit this paragraph whereas details may be stated in the conclusion section).

*We have rewritten this last paragraph to contextualize our work more explicitly on the methane paradox in the oxic waters of reservoirs, following the suggestions of the reviewer (please see page 3, Lines 84-92).*

The authors should, further, include an overview of reservoir characteristics (carbon/ phosphorus/nitrogen, trophic state, surface area, shoreline, mean depth, depth of the mixed layer). Also sampling locations should be characterized in a more detailed manner (location within the reservoir, shore distance, depth), for example, in tabular form. Towards the end of the Result&Discussion section I recommend recalling the general dispute between oxic and anoxic contribution and implementing the statistical results. Yet, there are not many literature information available, but please, place your overall finding into the context of DelSontro et al. 2018 and Günthel et al. 2020 (only studies so far presenting/summarizing contribution patterns for a series of lakes) who presented evidence for the importance of the oxic source increasing with basin size. Does the dataset indicate reservoir conditions favoring individual oxic methane sources? On some occasions the English should be improved throughout which will lead to a better understanding by the readership. Throughout the specific comments I acknowledged some of these occasions with rephrasing statements. Statistical correlations make up a huge part of the study. I would find it practical to include a table summarizing the outcome of all significant and non-significant correlations (stating type of test, predictor variables and effect, p-value; covering not only the GAM model results). It will be helpful, to reference display items like this table in the method section (in this regard please check all the Supplementary Materials).
Also, please include the correct units when statistical equations are stated throughout. Further, please check the wording when describing statistical results – on several occasions the statistical results are described with phrases like 'explains', 'determines' etc. Please use wording like 'points to explain', 'may determine', 'appears to be a main driver' etc. instead. Although the correlations appear to partially explain a lot of variance, no actual (production) rates have been measured, and this should be reflected by the language. The authors present a comprehensive dataset including many parameters. Please describe the obtained data in more detail throughout the

method section or when recalled by display items. While many of the conditions can be read out of other display items, it complicates the reading flow. For example, in Fig. 7, data points are grouped into stratified and mixed data. The reader has no information about corresponding depths. But different conditions in epilimnic versus hypolimnic waters may lead to different contribution patterns of individual methane sources. Did the authors check if splitting the dataset into epilimnic/hypolimnic data improves the correlations? Another example are the PCR results presented in Figs. S11 and S12. What reservoirs (depths, time points, replicates?) are resembled by samples 1-12? Please clarify these details.

*We have worked on the points indicated by the reviewer, including a new figure (Figure 1), and two new tables (Table 1 and 2) to describe the study reservoirs in the Methods section. We have also mentioned a new paper, recently published, of the same reservoirs and periods where there is more information on all these reservoirs (please see León-Palmero et al. 2020). We have also included a supplementary table summarizing the outcome of all significant and non-significant correlations (Supplementary Table 2). We have included the studies of DelSontro et al. (2018) and Günthel et al. (2020), and have revised the writing of the whole manuscript. We have fixed all the specific comments. Please see the comments below.*

SPECIFIC COMMENTS:
Abstract:
L 10-11. Please define CH4.

*We replaced "CH$_4$" by "Methane (CH$_4$)" (Line 10).*

L 15-16. Methane supersaturation is a common observation in aquatic systems, including oxic waters (e.g. Tang et al. 2016 and references herein). Without differentiating between oxic and anoxic methane sources this sentence adds little content (e.g. close to sediments/in anoxia methane supersaturation is generally expected). Rephrase or combine with the following sentence. I recommend avoiding the percentage numbers, especially if they do not refer to significant correlations with picoeukaryotes or cumulative chlorophyll-a.

*This sentence has been rewritten (Lines 16-17).*

L 16-17. Here, it is important to state the size of the study reservoirs.

*We have included the information of the size of the study reservoirs (Line 15).*

L 19. Replace 'determined' with a more appropriate wording, e.g. 'correlated with' etc.

*We have made the change suggested by the reviewer (Line 20).*

L 20-21. Please rephrase this sentence.

*We have rewritten that sentence (Lines 21 – 22).*

Introduction:

L 77-83. Explain your research questions in more detail and state why they are of interest (link them better to the previous introductory part). - Picoeukaryotes are first mentioned in the (too short) section describing research questions, not all readers are familiar with this terminology. Accordingly, it should be defined. What organism does it include? Why are they of interest within your research agenda - Actual methane production rates have not been measured but were deduced from proxies!

*As suggested by the reviewer, we have rewritten the last paragraph of the introduction in the revised manuscript (Lines 84 – 92).*
*We have described the photosynthetic picoeukaryotes and explained the groups found in the study reservoirs in the Results and Discussion section ("$CH_4$-production coupled to photosynthetic organisms").*

L 28. Change 'much' to 'more'
L 29. Change 'attributed to' to 'determined by'
L 41. Rephrase '$CH_4$ inputs may become from', e.g. '$CH_4$ may originate from'
L 62. Change 'contrary', e.g. to 'in contrast'

*We have made the corrections suggested by the reviewer in the manuscript (Lines 30, 30, 42, and 69).*

L 36-37. Add references, e.g. Tang et al. (2016) + references herein
L 44-47. Please incorporate the findings by Thalasso et al. (2020)
L 47-48. Please reference the findings by DelSontro et al. (2018)
L 52-53. Reference Hartmann et al. (2020), Bizic et al. (2020a) (isotopes); Khatun et al. (2020), Yao et al. (2016) (molecular approaches)

*We have included the references and the findings suggested by the reviewer (Lines 40, 48-51, 52-53, and 58-59).*

L 78. Diverse in what sense? Please clarify.

*The twelve reservoirs differ in size, morphometry, chemical, trophic, and watershed characteristics (more details in León-Palmero et al., 2020). We have included this reference in the text (lines 84-86).*
*In the revised manuscript, we have included two new tables to describe the study reservoirs. We show information about morphometry in Table 1 and information about the chemical and trophic characteristics in Table 2.*

Methods:
L 85-91. Please give an overview on the general reservoir characteristics (trophic state, size, location, temperature and oxygen conditions, epilimnion/hypolimnion depths etc). The reader requires this information to better place presented results into the study context. Searching for all parameters in various figures hampers a direct

comparison. In the following a series of different measurements are described. However, it is unclear if all these measurements were done simultaneously (at the same day or week) and at the same sites. Please clarify (maybe as a Supplementary Table). When measurements were done during mixed and stratified periods, have the exact same sites been re-sampled (location, depth, shore distance)?

*According to the reviewer comments, we have included a new figure with the geographical location of the reservoirs (Figure 1) and two new tables (Table 1 and 2) to describe the study reservoirs. Table 1 compiled the geographical coordinates, year of construction, and morphometric parameters of the reservoirs. In Table 2, we have included basic nutrient and trophic characteristics as carbon, phosphorus, nitrogen, and chlorophyll-a concentrations. We have also included a summary table with the main data in the supplementary material (Table S2). In this last table, we compiled the minimum, lower quartile, median, upper quartile, and maximum values for the dissolved $CH_4$ concentration ($\mu M$), saturation in $CH_4$ (%), water temperature (ºC), dissolved oxygen concentration (D.O., $\mu M$), oxygen saturation ($O_2$ saturation, %), the concentration of chlorophyll-a (Chl-a, $\mu g\ L^{-1}$), and the abundance of photosynthetic picoeukaryotes (PPEs, cell $mL^{-1}$) and cyanobacteria (CYA, cell $mL^{-1}$) in the mixing layer during the stratification period (epilimnion), and below the mixing layer during the stratification period, and in the mixing layer during the mixing period. Besides, we also deposited all the data that we used in this manuscript in the Pangaea database (https://doi.org/10.1594/PANGAEA.912535).*

L 88-90. Please rephrase this sentence.

*We have rewritten that sentence (Lines 103 - 105).*

L 91-93. Decide for either equation 1 or a definition in the running text to reduce redundancy.

*We have removed the description and kept the equation (1) (Line 107).*

L 102-104. Please state the units of each parameter as used throughout, and clarify what fluorescence relates to.

*We have included the units of each parameter in lines 118-120. We used the fluorescence to measure the in vivo concentration of chlorophyll-a. However, we did not use these data. To determine the concentration of chlorophyll-a we used the results of the pigment extraction (lines 177-181).*

L 104-105. Do you mean measurement intervals in 6 or 9 m steps? Please clarify.

*Based on the temperature and oxygen profiles, we selected from 6 to 9 depths for the discrete samplings of the water column. We tried to select the depths that best reflect the oxic, anoxic layers, and the transition between them, in the different reservoirs (Lines 120 – 121).*

L 148-152. Please clarify the difference between integrated mean (integrated over depth? Consider defining it as total amount/content.), and cumulative chlorophyll a concentration. Consider labeling them with different Symbols.

*We decided to remove the integrated mean of the chlorophyll-a of the Methods section to avoid misunderstandings, and because we did not use it in the Results section. We have included the equation for the cumulative chlorophyll-a calculation in the Method section (Lines 182-187).*

L 168. Please state the equations used to compute GPP, NEP and R.

*We have included the equations used to compute GPP, NEP and R in the Method section of the revised manuscript (Lines 199-232).*

L. 180-181. Please clarify what you mean by 'specific primers from similar studies'. What pure cultures did you use (type, culture conditions, origin)?

*We selected primers used in previous studies in freshwaters. We used the mcrA primers after West et al. (2012), who analyzed the abundance of the mcrA gene in lake waters and sediments. We used a culture of Methanosarcina acetivorans (ATCC 35395) as a positive control (Line 256). We used the primers for the gene phnJ from Yao et al. (2016), who analyzed the abundance of the gene phnJ in lake water. We used a culture of Rhodopseudomonas palustris (ATCC 33872) as a positive control (Line 263).*

*We bought both cultures from the Japan Collection of Microorganisms. Methanosarcina acetivorans (ATCC 35395) was bought as actively growing culture, while Rhodopseudomonas palustris (ATCC 33872) was bought as L-dried culture in an ampoule. We cultured the R. palustris on Luria-Bertani (LB) broth at 30 $^{o}$C.*

L 170-196. Please lay out when (in the context of other parameters) and where (location in the reservoir and depth) you sampled. Also, specify which reservoirs have been sampled (you state 12 reservoirs throughout section 2.1, but Fig. S11 only shows 10 samples). Please clarify, what samples have been measured (depth, mixed/stratified period). Also, please add this information to Fig. S11 and S12 (define 'samples 1-12'). Sample 6 in Fig. S12 appears to have a very weak signal close to the phnJ bar. Did you verify that this is no positive signal (e.g. edit light/contrast properties to better resolve this area)? What was used as negative controls (in both assays: mcrA, phnJ)?

*We clarified the details of the DNA analysis in the Method section "2.5 DNA analysis" in the revised manuscript (Lines 234 - 237). From the sampling of the water column, we selected 3 or 4 relevant depths for determining the abundance of the functional genes representing the epilimnion, metalimnion (oxycline), and hypolimnion/bottom layers during the stratification period. We also selected 3 or 4 similar depths during the mixing period. In total, we analyzed 77 samples: 41 samples from the stratification period, and 36 samples from the mixing period. However, we only included an electrophoresis gel per gene in the supplementary material of the manuscript to*

*simplify, and because all the result were negative. We analyzed the samples using PCR, and we also confirmed the negative results using quantitative PCR (qPCR).*

*In the Figure S11 (mcrA gene) we combined samples from stratification period (1-5) and mixing period (6-10) with peaks of dissolved $CH_4$ concentration. We have now included this information in the figure caption.*
*For the mcrA gene, the samples showed in the electrophoresis gel are:*
*1:  Colomera reservoir (depth= 6.5 m).*
*2: Negratín reservoir (16 m)*
*3: Los Bermejales reservoir (6m)*
*4:  Iznájar reservoir (4 m)*
*5: Francisco Abellán (16 m).*
*6: Iznájar reservoir (5 m).*
*7: Francisco Abellán reservoir (16 m).*
*8: San Clemente reservoir (12 m)*
*9: El Portillo reservoir (22 m)*
*10: Jándula reservoir (8m)*

*In Figure S12 (phnJ gene) we showed the samples 1:12, which correspond to the following samples from the mixing period. We have now included this information in the figure caption.*
*1: Cubillas reservoir (7.6 m)*
*2: Colomera reservoir (7 m)*
*3: Colomera reservoir (19 m)*
*4: Negratín reservoir (2 m)*
*5: Negratín reservoir (22 m)*
*6: Negratín reservoir (38 m)*
*7: La Bolera reservoir (12 m)*
*8: La Bolera reservoir (22 m)*
*9: Los Bermejales reservoir (6 m)*
*10: Los Bermejales reservoir (14 m)*
*11: Los Bermejales reservoir (30.5 m)*
*12: Iznájar reservoir (5m)*

L 194-196. What was the DNA range you investigated?

*The expected size of the PCR product was ~200 bp for the mcrA gene, and 400 bp for the phnJ gene. This information was included in the revised manuscript (Lines 256 and 263).*

Results & DISCUSSION:
L 218-219. Please avoid the large and unpractical percentage numbers (hard to read). Given, that the authors state the dissolved concentration, no content is lost when removing these numbers. For example, the authors could incorporate the average saturation concentrations in the following sentence.

*We have removed these large number of % supersaturation and these data are reported in Supplementary Table S1.*

L 223-225. The references are mixed up. E.g., Donis et al., Grossart et al., Tang et al. investigated temperate lakes, but Murase et al. researched a tropical lake etc. Please rephrase this sentence.

*We thank this observation from the reviewer. We have rewritten this sentence (Lines 293-295).*

L 225. Please define the depth of 'surface waters'. Why is emphasize placed on Lake Kivu (not listed throughout the previous sentence)?

*We have changed "surface waters" by "surface mixing layer during the stratification period (i.e., epilimnion)" (Line 295).*

L 248. Please mention that the literature refers to studies in lakes. Note, a metalimnic methane maximum can be controlled by physical (e.g. differential gas solubility due to temperature change, emission) and biological factors (e.g. light inhibition of methane oxidation, variable phytoplankton methane production due to availability of nutrients/light/precursors). Also note, Kathun et al. (2019) presented a series of lakes with and without metalimnic methane maxima.

*We have included the results of Kathun et al (2019) and the further explanations of the metalimnetic $CH_4$ maximum in the revised manuscript (Lines 319-326).*

L 254. Referenced literature is not about boreal lakes.

*Reviewer was right and we have changed this sentence (Line 330).*

L 256-263. Reference Tang et al. (2014) who also presented a distinction between oxic and anoxic methane concentrations in oxic and anoxic lake waters.

*We have included the reference suggested by the reviewer (Line 333).*

L 264-269. Please indicate what depths you analyzed.

*As we have explained in the method section, we selected 3 or 4 depths for determining the abundance of the functional genes representive of the epilimnion, metalimnion (oxycline), and hypolimnion/bottom layers during the stratification period. We also selected 3 or 4 similar depths during the mixing period. In total we have analyzed 77 samples: 41 samples from the stratification period, and 36 samples from the mixing period. From these 77 samples selected, twelve samples belonged to the anoxic group (D.O. < 7.5 µM). We have included this information this in the revised manuscript (Line 343).*

L 272-273. Please rephrase the sentence. Also, please emphasize archaeal methane

production is absent in the anoxic water column.

*We have rewritten this paragraph and have made explicit that archaeal methane production is absent in the water column (lines 349-351).*

L 282. Please reword 'depend'. E.g. correlate with.

*We have changed the word "depend" for "was correlated to" (Line 360).*

L 289-291. Please reference these considerations/results here.

*We have rewritten these lines (371-375).*

L 296. Please define 'extreme' P limitation. Different Organisms require different amount of P. Accordingly, this is a relative terminology. Does it relate to the N:P ratio? Please clarify.
and
L 341-342. Define 'extreme' limitation. According to my knowledge, it is unknown what are the P levels (N:P ratio) triggering MPN degradation and corresponding methane production in the field.

*To avoid misunderstanding, we have changed the word "extreme" by "severe". P-limitation was defined based on the ratio between the dissolved inorganic nitrogen (DIN) and total phosphorus (TP) that is widely used in the scientific literature (e.g. Morris and Lewis 1988). DIN:TP ratios > 4 are indicative of P-limitation (Axler et al 1994). All the study reservoirs have DIN:TP ratio larger than 15. However, despite this assumed P-limitation, we did not observe a relationship with CH4 and we were unable to detect the phnJ gene.*

L 299-304. I appreciate the authors approach to evaluate the contribution of anoxic methane based on morphometrical parameters. However, lateral transport which is seen as major source of epilimnic methane is modulated by the shore-mid distance. Accordingly, when correlations are done this should be accounted for. Please clarify shore-mid distances in your dataset. I think, Hofmann et al. 2010 and DelSontro et al. 2018 should be referenced as these studies show major contributions by the littoral (especially in smaller lakes).
And L 313-314. Please discuss why there was no significant correlation between dissolved methane and shallowness index. Potential reasons might be: variable sediment methane production rates among reservoirs (variable among temperature, trophic state, sediment porosity, soil type and community etc.) the ratio between oxic and anoxic methane production may lead to distortion. The authors may have other points. Did you try the ratio of 'mean depth: depth of surface mixed layer' as a proxy for lateral input (depth of the mixed layer relates to the amount of temperate sediments)?
L 317-318. Please rephrase.

*We sampled the water column near the dam, in the open waters of the reservoir at the*

*same location during the stratification and the mixing period. Unfortunately, we did not measure the shore-mid distances from the sampling point. For this reason, we used the shallowness index as a proxy for the lateral transport, but we did not find significant results between this index and the dissolved $CH_4$ in oxic waters. We also analyzed the relationship between the shallowness index and the dissolved $CH_4$ in the mixing layer during the stratification period (i.e., epilimnion), but the result was still not significant (p-value = 0.109). We consider that the lateral transport from lateral zones may be important in areas closed to the littoral zone, but not in the open waters of the study reservoirs likely because all the reservoirs have a size larger than 1Km2*

*We have extended and rewritten the discussion on this issue. We have also included the references suggested by the reviewer (Lines 398-408).*

*We also analyzed the relationship between the ratio of 'mean depth: depth of surface mixed layer' as a proxy for lateral input, and the concentration of $CH_4$ in the oxic samples during the stratification period and the relationship was not significant (p-value = 0.676). We also tried the relationship between this ratio and the concentration of $CH_4$ in the mixing layer during the stratification period, and the relationship was not significant (p-value = 0.896).*

L 304-305. Please reference the lateral transport model by DelSontro et al. (2018) which agrees with the observation of exponential decay functions. A later discussion on this decay function should be warrant. E.g., close to the shore there should be a high content of dissolved methane that originated from the sediments; with increasing distance from the shore the relative contribution of the oxic methane source should increase.

*At this point, we disagree with the reviewer. DelSontro et al. (2018) described exponential decay functions relating dissolved $CH_4$ with the distance to the shoreline. In contrast, in this study, we found exponential decay functions that related the dissolved $CH_4$ to the mean depth of the system. This fact could be related to a promotion of diffusion from hypolimnion associated with a decline in the hydrostatics pressure. We have discussed this issue in this new version (Lines 396-397).*

L 306-307. Concentrations increased exponentially versus what parameter?

*The dissolved $CH_4$ concentration increased exponentially in reservoirs with a mean depth shallower than 16 meters (Lines 391-392).*

L 315-317. I think the authors should remove the wind speed correlation throughout. Wind-forcing, or more generally, turbulence which drives surface emission and substantially affects surface water methane concentrations, is modulated by many environmental factors (basin geomorphometry, temperature, rain etc.). Also, internal production rates, lateral methane input and methane oxidation modulate surface concentrations beside emission. Accordingly, correlating wind speed with surface concentrations is an over-simplification and might be misleading.

We agree with the reviewer, so we have removed the wind speed correlation.

L 318-319. Please define what you mean by 'extreme supersaturation' and where these concentrations were found.

*We have rewritten these lines in the revised manuscript (Lines 403 – 408).*

L 323-327. This is an interesting observation. Consider highlighting this observation more. Is it possible that you filtered off the attachment partners of methanogenic Archaea (you filtered water samples before molecular analyses)?

*For the molecular analysis, to keep clean the procedure, we pre-filtered the water through 3.0 µm pore-size filters. Then, we think that if methanogenic archaea were associated with particles or organisms larger than 3.0 µm, we were not able to detect them. We have made explicit this issue in the new version (Lines 413-420).*

L 342-343. Are these correlations simple x-y regressions or do they include for multiple predictor variables?

*They were simple x-y regressions. We have included these results in a new Table S2.*

L 345-350. Fig S12. Can you change the picture properties (light, contrast); it seems, there is a faint bluer in sample 6. Cyanobacteria have been detected (e.g. Fig. S13). What type of cyanobacteria were present? Do these result agree with findings by Yao et al. 2016 (some types possess the methane generating enzyme machinery)?

*Besides the PCR analysis, we also confirmed the negative results using quantitative PCR. We have included more details in Figure S12.*
*We quantified cyanobacteria only using flow cytometry. Therefore, unfortunately, we cannot provide the taxonomical identification. We identified them mainly as phycoerythrin-rich picocyanobacteria, although we also detected phycocyanin-rich picocyanobacteria in Béznar reservoir. They probably belong to the non-marine Synechococcus/Cyanobium clade, that has been isolated from lakes from a wide range of trophic states and geographical regions* (Callieri et al., 2013), *but we did not analyze them taxonomically.*

L 352-364. Many information listed belong to the result/method section. Please remove redundancy.

*We have removed the redundant sentences.*

L 365-366. Given that many factors may affect the dissolved methane concentration (as discussed throughout) a p-value of 0.077 might still point to a connection. It has been stated n=12; does that mean 1 value per reservoir during the stratified period? Same questions on other occasions. Please clarify.

*Indeed, we only obtained just one value of GPP and NEP per reservoir during the*

*stratification period (i.e., n=12). We have mentioned this information in the new version (Line 446). We have also included the fact that the relationship between GPP and dissolved CH$_4$ was marginally significant (Lines 458-459, Table 3).*

383-384. Please add the reference Hartmann et al. 2020 who presented methane production by green algae, cyanobacteria, cryptophytes and diatoms.

*We have included this reference in the revised manuscript (Line 487).*

L 396-405. Please mention methylated amines which can also serve as methane precursors (e.g. Bizic et al. 2018, Bizic et al. 2020b).

*We have included these references in the revised manuscript (Line 499-500).*

L 412-414. Please emphasize that this cyanobacterial methane 'production' does here not relate to MPN degradation (following the absence of phnJ).

*We have explicitly included the there are other methylated by-products different from MPN in the revised manuscript (Line 486).*

L 435-436. Please rephrase and avoid the percentage number. Results by Bogard et al. (2014), Donis et al (2017), DelSontro et al. (2018) and Günthel et al. (2019) suggest that in larger waterbodies the majority of surface mid-water methane (/emission) might be explained by the oxic source.

*We have rewritten this paragraph and included the references suggested by the reviewer (Lines 540-544).*

L 437-438. Please rephrase.

*We have deleted that sentence.*

L 458-460. I suggest moving the PPEs description to the result section or even earlier, when the terminology PPE is defined.

*We have partially moved the description of the PPEs (lines 447- 451).*

L 461-465. Please indicate to what extent the different methane sources might explain the dissolved methane concentrations (e.g. in percentage – relates to Fig. 8a).
and
L 784. (corresponding to Fig. 8a). I am not sure, 'significance' is the best terminology at this point. Consider rephrasing y-axes to explanatory power. Could the author please comment on the partial effect of 'sediment methane production'
(Fig. 8b)? Why is the trendline reversing the slope after ca. 27.5m mean depth? Does this parameter indicate bigger (deeper) reservoirs than 2s7.5 m mean depth have a higher sediment contribution to mid-water methane?

*Using the GAM models, we can assess the relative **significance** of smooth terms of the explanatory variables, and the total deviance and variance explained by the model. The significance of the smooth terms shows how significant the smooth terms of the model are. The F-tests on smooth terms (rather than for the full model) are joint tests for equality to zero for all of the coefficients making up a single spline term. F-test is the ratio of the explained and unexplained variance. The model includes various terms: four variables for the stratification period, and two for mixing period. Each of these terms is more or less important explaining the variance of the response variable "dissolved $CH_4$ concentration". Some terms have a high significance (high F value, and small p-value), as the PPEs abundance for the mean depth, and that means that most probably, in reality, the PPEs and the mean depth are significant factors contributing to the dissolved $CH_4$ concentration. We studied the contribution of each term by comparing F values in Fig 9a and Fig10a in the revised manuscript.*

*Unfortunately, we did not measure the $CH_4$ production in the sediments or in situ in the oxic zone by PPEs or Cyanobacteria. Therefore, we cannot account for the exact amount of $CH_4$ that comes from each source.*

*The partial effect of 'sediment methane production' (mean depth as a surrogate) was different during the stratification and the mixing period. We have included an explanation of these differences in the revised manuscript (lines 520- 523).*

L 467-468. Given other work on relationships between methane and chlorophyll/noncyanobacteria phytoplankton (Tang et al. 2016, Hartmann et al. 2020), I think novel is not the right terminology here.

*We have deleted this sentence.*

L 475-476. Please rephrase.

*We have rewritten these sentences rephrased that section (Lines 576-577).*

*We have included the regression statistics in the figure legends, as suggested by the reviewer.*

L 769-771. Did you have data about reactive phosphorus? Using biologically accessible phosphorus instead of total phosphorus might improve the correlation statistics.

*In the study reservoir, we have measured the total phosphorus concentration (TP), the total dissolved phosphorus concentration (TDP), and the soluble reactive phosphorus (SRP). We also studied the relationship between the DIN: SRP ratio and the dissolved $CH_4$ concentration in oxic waters, but it was not significant during the stratification period  (p-value: 0.195) and the mixing period (p-value: 0.153). We decided to include just TP to reduce the manuscript since all the results were negative.*

L 775-780. In case R2<1.0, the functions do not entirely explain the dissolved methane concentration data but only a certain fraction of data variance (e.g. 40% at R2=0.40 in a simple regression). Please rephrase accordingly. What water depths do these readings (n=78 or 82) correspond to? In case some belong to the epilimnion and some to the hypolimnion, different nutrient availabilities may affect the correlation statistics and mask potential relationships.

*We have rewritten this figure caption.*

*Throughout the section "CH$_4$ sources in oxic waters" we studied the samples with dissolved oxygen concentration higher than 7.5 µM (n = 160, oxic samples). These samples belonged to the stratification period (n = 78) and the mixing period (n = 82). Therefore, the samples from the stratification period contained samples from both layers: epilimnion and hypolimnion.*

*We do not consider that different nutrient availabilities may affect the results in the epilimnion and the hypolimnion because the more significant differences in nutrient concentrations (N, C, P) were found among reservoirs, not among the depths of the same reservoir. We performed a Kruskal Wallis test (for non normally distributed data) on the dissolved organic carbon (DOC) concentration (µM), dissolved inorganic nitrogen (DIN) concentration (µM), total phosphorus (TP) concentration (µM), and on the DIN:TP molar ratio to test the differences among reservoirs and among depths during the stratification period. We did not test the differences during the mixing period because the water column is completely mixed. We recorded the p-values in the following table:*

|  | *Differences among reservoirs* | *Differences among depths* |
|---|---|---|
| *DOC* | *< 0.01* | *0.111* |

| | | |
|---|---|---|
| DIN | < 0.01 | 0.100 |
| TP | < 0.001 | 0.211 |
| DIN:TP | < 0.001 | 0.763 |

Supplementary Materials:

Table S1. Please discuss why the statistical correlation leads to substantial differences when applied to the combined data set of stratified and mixed period (e.g. no significant correlation listed with mean depth what is here used as a proxy for conventional sediment methanogenesis). Please discuss these details in the main document.

*The Table S3 we have summarized the results for the GAM model during the stratification period, the mixing period, and all the dataset combined. In this new version, we have included a GAM model using all the dataset and variables widely used in limnology (temperature, chlorophyll-a, and mean depth) for potential future use that is included in the main text (Lines 528-532).*

**References:**

[revised manuscript text omitted]